# Can AI-Generated Text be Reliably Detected?
# Stress Testing AI Text Detectors Under Various Attacks

**Vinu Sankar Sadasivan** *vinu@umd.edu*
*Department of Computer Science, University of Maryland*

**Aounon Kumar** *aokumar@hbs.edu*
*Department of Computer Science, Harvard University*

**Sriram Balasubramanian** *sriramb@umd.edu*
*Department of Computer Science, University of Maryland*

**Wenxiao Wang** *wwx@umd.edu*
*Department of Computer Science, University of Maryland*

**Soheil Feizi** *sfeizi@umd.edu*
*Department of Computer Science, University of Maryland*

**Reviewed on OpenReview:** *https://openreview.net/forum?id=OOgsAZdFOt*

## Abstract

Large Language Models (LLMs) can perform impressively well in various applications, such as document completion and question-answering. However, the potential for misuse of these models in activities such as plagiarism, generating fake news, and spamming has raised concerns about their responsible use. Consequently, the reliable detection of AI-generated text has become a critical area of research. Recent works have attempted to address this challenge through various methods, including the identification of model signatures in generated text outputs and the application of watermarking techniques to detect AI-generated text. These detectors have shown to be effective under their specific settings. In this paper, we stress-test the robustness of these AI text detectors in the presence of an attacker. We introduce *recursive paraphrasing* attack to stress test a wide range of detection schemes, including the ones using the watermarking as well as neural network-based detectors, zero-shot classifiers, and retrieval-based detectors. Our experiments conducted on passages, each approximately 300 tokens long, reveal the varying sensitivities of these detectors to our attacks. We also observe that these paraphrasing attacks add slight degradation to the text quality. We analyze the trade-offs between our attack strength and the resulting text quality, measured through human studies, perplexity scores, and accuracy on text benchmarks. Our findings indicate that while our recursive paraphrasing method can significantly reduce detection rates, it only slightly degrades text quality in many cases, highlighting potential vulnerabilities in current detection systems in the presence of an attacker. Additionally, we investigate the susceptibility of watermarked LLMs to spoofing attacks aimed at misclassifying human-written text as AI-generated. We demonstrate that an attacker can infer hidden AI text signatures without white-box access to the detection method, potentially leading to reputational risks for LLM developers. Finally, we provide a theoretical framework connecting the AUROC of the best possible detector to the Total Variation distance between human and AI text distributions. This analysis offers insights into the fundamental challenges of reliable detection as language models continue to advance. Our code is publicly available at https://github.com/vinusankars/Reliability-of-AI-text-detectors.

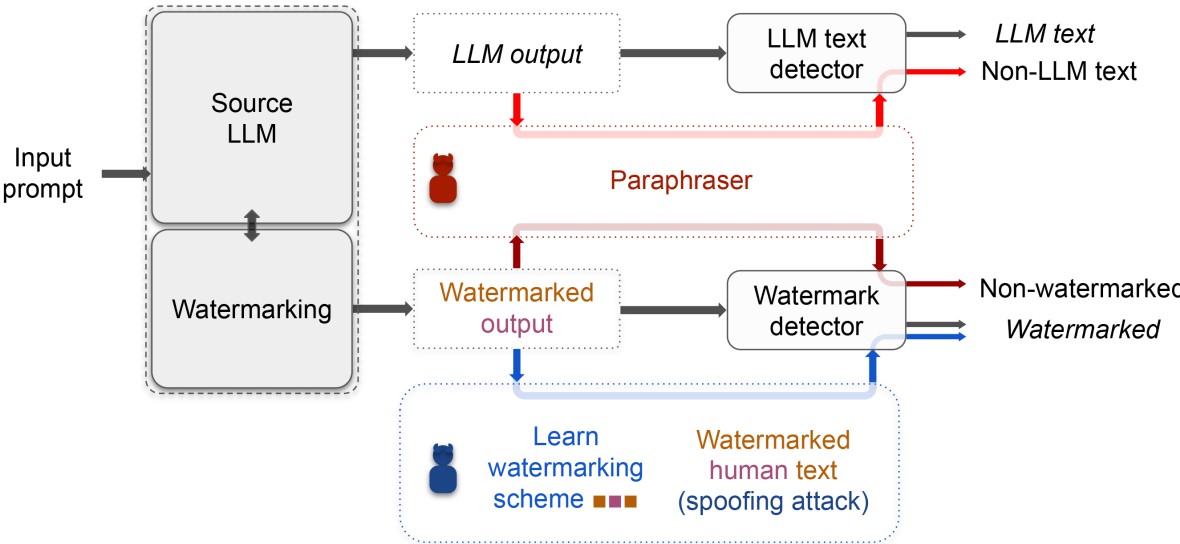

Figure 1: An illustration of vulnerabilities of existing AI-text detectors. We consider both watermarking-based and non-watermarking-based detectors and show that they are not reliable in practical scenarios. Colored arrow paths show the potential pipelines for adversaries to avoid detection. In red: an attacker can use a paraphraser to remove the LLM signatures from an AI-generated text to avoid detection. In blue: an adversary can query the watermarked LLM multiple times to learn its watermarking scheme. This information can be used to spoof the watermark detector.

## 1 Introduction

Artificial Intelligence (AI) has made tremendous advances in recent years, from generative models in computer vision (Rombach et al., 2022; Saharia et al., 2022) to generative models in natural language processing (NLP) (Brown et al., 2020; Zhang et al., 2022; Raffel et al., 2019). Large Language Models (LLMs) can now generate texts of supreme quality with the potential in many applications. For example, the recent model of ChatGPT (OpenAI, 2022) can generate human-like texts for various tasks such as writing codes for computer programs, lyrics for songs, completing documents, and question answering; its applications are endless. The trend in NLP shows that these LLMs will even get better with time. However, this comes with a significant challenge in terms of authenticity and regulations. AI tools have the potential to be misused by users for unethical purposes such as plagiarism, generating fake news, spamming, generating fake product reviews, and manipulating web content for social engineering in ways that can have negative impacts on society (Adelani et al., 2020; Weiss, 2019). Some news articles rewritten by AI have led to many fundamental errors in them (Christian, 2023). Hence, there is a need to ensure the responsible use of these generative AI tools. In order to aid this, a lot of recent research focuses on detecting AI-generated texts.

Recent works propose several ways, such as using trained neural network-based detectors, zero-shot detectors, watermarking, and retrieval-based detectors for flagging AI-generated text. These detectors, especially watermarking, have shown to be effective in various settings. Neural network-based detectors approach the detection problem as a binary classification task (OpenAI, 2019; Jawahar et al., 2020; Mitchell et al., 2023; Bakhtin et al., 2019; Fagni et al., 2020; Li et al., 2024). For example, OpenAI fine-tunes RoBERTa-based (Liu et al., 2019) GPT-2 detector models to distinguish between non-AI generated and GPT-2 generated texts (OpenAI, 2019). This requires such a detector to be fine-tuned with supervision on each newly released LLM for reliable detection. Zero-shot detectors address this downside of trained detectors by performing the detection task without any additional training overhead (Solaiman et al., 2019; Ippolito et al., 2019; Gehrmann et al., 2019). These works evaluate the expected per-token log probability of texts and perform thresholding to detect AI-generated texts. Mitchell et al. (2023) observe that AI-generated passages tend to

lie in negative curvature of log probability of texts. To leverage this observation, they propose DetectGPT, a zero-shot LLM text detection method. However, zero-shot detectors require access to the original model used to generate the AI text to achieve the best performance. Additionally, neural network-based and zero-shot detectors rely on a deep net for their detection, and they can be vulnerable to adversarial and poisoning attacks (Goodfellow et al., 2014; Sadasivan et al., 2023; Kumar et al., 2022; Wang et al., 2022). In comparison to these methods, watermarking significantly eases the detection of AI-generated text by imprinting specific patterns on them that are imperceptible to humans (Atallah et al., 2001; Wilson et al., 2014; Kirchenbauer et al., 2023a; Zhao et al., 2023b; Kuditipudi et al., 2024; Zhao et al., 2023a). Soft watermarking proposed in Kirchenbauer et al. (2023a) partitions tokens into "green" and "red" lists, as they define, to help create these patterns. A watermarked LLM samples a token, with high probability, from the green list determined by a pseudo-random generator seeded by its prefix token. The watermarking detector would classify a passage with a large number of tokens from the green list as AI-generated. The soft watermarking approach of Kirchenbauer et al. (2023a) has been shown to be effective in various settings and remains one of the popular approaches for detecting AI-generated text. For example, their watermarking scheme can achieve a high true positive rate of 99.8% at a false positive rate of 1% on a task for classifying AI text against human-written news articles. However, for watermarking to be an effective tool for preventing AI misuse, it has to be enforced on all the major LLM generators. Otherwise, an adversary could use a non-watermarking LLM for their purposes. Krishna et al. (2023) introduces an information retrieval-based detector by storing the outputs of the LLM in a database. For a candidate passage, their algorithm searches this database for semantically similar matches for detection. However, storing user-LLM conversations might cause serious privacy concerns.

Several recent news (Fowler, 2023; Hill, 2022; Das, 2023; Quach, 2023; Al-Sibai, 2023) show that some of these popular AI-text detectors fail in practical settings. In this paper, through several experiments, we stress-test state-of-the-art AI-text detectors to evaluate their robustness in the presence of an attacker (Wolff, 2020; Aaronson, 2022; Liang et al., 2023; Pu et al., 2023; Wang et al., 2023). In §2, we have developed a *recursive paraphrasing attack* that use neural network-based paraphrasing to recursively paraphrase the source LLM's output text. We perform experiments with our automated recursive paraphrasing to show the sensitivity of a range of AI text detectors to *type-II error* (detecting AI text as human text). For instance, **recursive paraphrasing attack on watermarked texts, even over relatively long passages of 300 tokens in length, can drop the detection rate (true positive rate at 1% false positive rate or TPR@1%FPR) from 99.3% to 9.7%.** We find that our attack can add slight degradations to the AI text quality. Hence, we analyze the trade-offs between our attack and the resulting text quality, measured through human studies, perplexity scores, and accuracy of text benchmarks. Our attack differs from the relatively weaker attack from Kirchenbauer et al. (2023a) where they perform span replacement by replacing random tokens (in-place) using an LLM. Thus, our experiments show the sensitivity of the watermarking scheme against paraphrasing attacks in the presence of a stronger attacker. Zhang et al. (2024) and Lu et al. (2023) are also substitution-based attacks. Zhang et al. (2024) have a different attack objective when compared to our attack. Their attack is performed to maintain the quality of the text alone and not semantic similarity. Hence, their attack can change the original content. Lu et al. (2023) employs in-context optimization through iterative generation of word- or sentence-level substitutions using a proxy AI text detector to guide the substitutions. This positions their attack as an adversarial algorithm for evading text detection. In contrast, our approach focuses on non-adversarial iterative or recursive text paraphrasing attacks.

After paraphrasing, the area under the receiver operating characteristic (AUROC) curves of zero-shot detectors (Mitchell et al., 2023) drops from 96.5% to 25.2%. We also observe that the performance of neural network-based trained detectors (OpenAI, 2019) deteriorates significantly after our paraphrasing attack. For instance, the TPR@1%FPR of the RoBERTa-Large-Detector from OpenAI drops from 100% to 60% after paraphrasing. In addition, we show that the retrieval-based detector by Krishna et al. (2023) designed to evade paraphrase attacks is vulnerable to our recursive paraphrasing. In fact, the accuracy of their detector falls from 100% to below 60% with our recursive paraphrase attack.

To quantify the drop in text quality after recursive paraphrasing, we perform MTurk human evaluation studies and measure other automated metrics such as perplexity and text benchmark accuracy. **Our human evaluation study shows that 77% of the recursively paraphrased passages are rated high quality**

**in terms of content preservation, and** $89\%$ **of them are rated high quality in terms of grammar or text quality.** We also show that our **recursive paraphrasing, when applied to a text benchmark such as a question-answering dataset, does not affect the performance**, providing additional evidence that recursive paraphrasing does not hurt the content of the original text. Though an attacker may further improve the text quality with manual interventions, paraphrasing attacks can be sufficient for an adversary to perform social engineering tasks such as spamming, phishing, or spreading propaganda.

Moreover, we show the possibility of **spoofing attacks** on various AI text detectors in §3. In this setting, an attacker generates a non-AI text that is detected to be AI-generated, thus adversarially increasing *type-I error* (falsely detecting human text as AI text). An adversary can potentially launch spoofing attacks to produce derogatory texts that are detected to be AI-generated to affect the reputation of the target LLM's developers. In particular, we show that an adversary can infer hidden AI text signatures without having white-box access to the detection method. For example, though the pseudo-random generator used for generating watermarked text is private, we develop an attack that adaptively queries the target LLM multiple times to learn its watermarking scheme. An *adversarial human* can then use this information to compose texts that are detected to be watermarked. Figure 1 shows an illustration of some of the vulnerabilities of the existing AI-text detectors. Gu et al. (2024) build upon our spoofing attacks by employing watermarked data distillation to train a student model to replicate the next-token distribution.

Finally, in §4, we present a theoretical result regarding the hardness of AI-text detection. Our main result in Theorem 1 states that the AUROC of the best possible detector differentiating two distributions $\mathcal{H}$ (e.g., human text) and $\mathcal{M}$ (e.g., AI-generated text) reduces as the total variation distance $\mathsf{TV}(\mathcal{M}, \mathcal{H})$ between them decreases. Note that this result is true for any two arbitrary distributions $\mathcal{H}$ and $\mathcal{M}$. For example, $\mathcal{H}$ could be the text distribution for a person or group and $\mathcal{M}$ could be the output text distribution of a general LLM or an LLM trained by an adversary to mimic the text of a particular set of people. Essentially, adversaries can train LLMs to mimic human text as they get more sophisticated, potentially reducing the TV distance between human and AI text, leading to an increasingly more difficult detection problem according to our Theorem 1. Although estimating the exact TV between text distributions from a finite set of samples is a challenging problem, we provide some empirical evidence, over simulated data or via TV estimations, showing that more advanced LLMs can potentially lead to smaller TV distances. Thus, **our Theorem 1 would indicate an increasingly more difficult reliable detection problem** in such cases. Our theory also indicates that if a detector becomes more robust to type-I errors, type-II errors will increase, revealing a fundamental tradeoff between type-I and type-II errors for the AI text detection problem. Similar tradeoffs have been explored in other domains as well. For example, Khajavi & Kuh (2016) study the relationship between detection performance and KL divergence between input distributions in the context of covariance selection. Thapliyal & Hwang (2022) show that undetectable cyberattacks can be generated by mimicking the input-output data distribution of network control systems. Although not a surprising result, Theorem 1 is the first to link this tradeoff to the detection of AI-generated content to our knowledge.

Identifying AI-generated text is a critical problem to avoid its misuse by users for unethical purposes such as plagiarism, generating fake news, and spamming. However, blindly relying on these detectors may *not* be the right solution to tackle this issue since it can cause its own damages, such as falsely accusing a human of plagiarism. Our results highlight the sensitivities of a wide range of detectors to both evasion and spoofing attacks and indicate the difficulty of developing reliable detectors in the presence of an attacker. We hope to reveal the sensitivity of AI text detectors to various attacks in our stress tests experiments.

In summary, we make the following contributions in this work.

- Our work is the *first to comprehensively analyze* the robustness of four different classes of detectors, including watermarking-based, neural network-based, zero-shot, and retrieval-based detectors, and stress-test them in the presence of an attacker (in §2). In particular, the *recursive paraphrasing attack* that we develop is the first method that can break watermarking (Kirchenbauer et al., 2023a) and retrieval-based (Krishna et al., 2023) detectors. We also provide experiments to analyze the resulting text quality after our attack to find that recursive paraphrasing only leads to a slight text quality degradation in many cases.

| ppi | | i=1 | i=2 | i=3 | i=4 | i=5 | All ppi |
|---|---|---|---|---|---|---|---|
| Content preservation | Avg. rating | $4.0 \pm 0.8$ | $4.1 \pm 0.8$ | $3.9 \pm 0.9$ | $4.2 \pm 0.9$ | $3.7 \pm 1.1$ | $4.0 \pm 0.9$ |
| | Ratings 5&4 | 70.2% | 77.2% | 63.2% | 80.0% | 61.4% | 70.4% |
| Grammar or text quality | Avg. rating | $4.28 \pm 0.67$ | $4.12 \pm 0.50$ | $4.12 \pm 0.53$ | $4.11 \pm 0.64$ | $4.07 \pm 0.53$ | $4.14 \pm 0.58$ |
| | Ratings 5&4 | 87.72% | 92.98% | 91.23% | 84.21% | 89.47% | 89.12% |

Table 1: Summary of the MTurk human evaluation study on content preservation and grammar or text quality of the recursive paraphrases with DIPPER that we use for our attacks. Ratings are on a Likert scale of 1 to 5. See Appendix B.1 for details.

| ppi | | i=1 | i=2 | i=3 | i=4 | i=5 | All ppi |
|---|---|---|---|---|---|---|---|
| Content preservation | Avg. rating | $4.37 \pm 0.63$ | $4.18 \pm 0.67$ | $3.93 \pm 0.71$ | $3.9 \pm 0.75$ | $3.85 \pm 0.78$ | $4.05 \pm 0.2$ |
| | Ratings 5&4 | 91.67% | 85.0% | 80.0% | 78.3% | 80.0% | 83.0% |
| Grammar or text quality | Avg. rating | $4.62 \pm 0.55$ | $4.28 \pm 0.73$ | $4.26 \pm 0.65$ | $4.22 \pm 0.64$ | $4.17 \pm 0.74$ | $4.31 \pm 0.35$ |
| | Ratings 5&4 | 96.67% | 83.33% | 88.33% | 88.3% | 83.33% | 88.0% |

Table 2: Summary of the MTurk human evaluation study of the recursive paraphrases with LLaMA-2-7B-Chat.

- Our work is the first to show that existing detectors are vulnerable against *spoofing attacks* where an adversarial human aims to write a (potentially derogatory) passage falsely detected as AI-generated *without* having a white-box access to the detection methods (in §3). For instance, as proof of concept, we show that an adversary can infer the watermarking signatures by probing the watermarked LLM and analyzing the statistics of the generated tokens.

- Our work is the first to establish a theoretical connection between the AUROC of the best possible detector and the TV distance between human and AI-text distributions that can be used to study the hardness of the reliable text detection problem (in §4). Our theory also reveals a fundamental tradeoff between type-I and type-II errors for the AI text detection problem.

## 2 Evading AI-Detectors using Paraphrasing Attacks

In this section, we first present the experimental setup for our paraphrasing attacks in §2.1. We also provide experiments in §2.2 to study the trade-off between evasion success and text quality after the attack. §2.3 and §2.4 show the effect of the paraphrasing attacks on watermarking and non-watermarking detectors, respectively. In Appendix A.1, we provide attack experiments with Llama-2-13B as the target model on additional detectors.

### 2.1 Attack Setup and Paraphrasing Methods

For evasion attacks, we consider a scenario where an adversary takes an AI response generated by a target model and then modifies the AI response in an automated and scalable fashion to evade detection. In this work, we propose the adversary modify the AI text from model $\mathcal{L}$ using an AI paraphraser $\mathcal{P}$ to evade detection. Note that the adversary might be incentivized to use a detectable model $\mathcal{L}$ (say, watermarked) if $\mathcal{L}$ is powerful or might have been fine-tuned for specific applications. In these cases where $\mathcal{L}$ could answer a user prompt better, an adversary would still prefer to use the watermarked $\mathcal{L}$ to generate a response and then use a less detectable AI model $\mathcal{P}$ for paraphrasing to evade detection. We quantify the text quality using automated metrics such as perplexity and human studies. As shown in §2.2, our evasion attacks only lead to a slight degradation in text quality while successfully evading detectors most of the time. Note that the amount of degradation that can be tolerated is application-specific. For example, an adversary could tolerate more quality degradation when generating a social media post than when generating a news article.

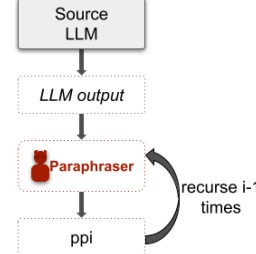

Figure 2: Recursive paraphrasing

We use the "document" features of the XSum dataset (Narayan et al., 2018) containing 1000 long news articles (∼300 tokens in length) for our experiments. In Appendix A, we perform experiments with additional datasets – a medical text dataset PubMedQA (Jin et al., 2019) and a dataset with articles from 10 different domains Kafkai (Kafkai, 2020). As target

| Paraphraser | Evaluation | AI text | pp1 | pp2 | pp3 | pp4 | pp5 |
|---|---|---|---|---|---|---|---|
| DIPPER | Mean perplexity | 5.2 | 7.7 | 7.8 | 8.5 | 7.7 | 8.7 |
| | QA performance | 97% | 97% | 96% | 96% | 96% | 95.5% |
| LLaMA-2-7B-Chat | Mean perplexity | 5.2 | 8.1 | 9.3 | 9.0 | 10.3 | 10.5 |
| | QA Performance | 97% | 97% | 97% | 97% | 97% | 97% |

Table 3: Automated evaluation of the text quality of recursive paraphrases using perplexity measures with respect to OPT-13B and question-answering benchmark accuracy.

LLMs, we use OPT-1.3B and OPT-13B (Zhang et al., 2022) language models with 1.3B and 13B parameters, respectively. In Appendix A, we also evaluate our attacks with GPT-2 Medium (Radford et al., 2019) as the target model. We use three different neural network-based paraphrasers – DIPPER with 11B parameters (Krishna et al., 2023), LLaMA-2-7B-Chat with 7B parameters (Touvron et al., 2023), and T5-based paraphraser (Damodaran, 2021) with 222M parameters. Suppose a passage $S = (s_1, s_2, ..., s_n)$ where $s_i$ is the $i^{th}$ sentence. DIPPER and LLaMA-2-7B-Chat paraphrase $S$ to be $S' = f_{strong}(S)$ in one-shot while the light-weight T5-based paraphraser would output $S' = (f_{weak}(s_1), f_{weak}(s_2), ..., f_{weak}(s_n))$ where they can only paraphrase sentence-by-sentence. DIPPER and LLaMA-2-7B-Chat also have the ability to input a context prompt text $C$ to generate higher-quality paraphrasing $S' = f_{strong}(S, C)$. We can also vary two different hyperparameters of DIPPER to generate a diverse number of paraphrases for a single input passage.

We use DIPPER and LLaMA-2-7B-Chat for recursive paraphrasing attacks since they provide high-quality paraphrasing when compared to the 222M parameter T5 model. Let an LLM $\mathcal{L}$ generate AI text output $S = \mathcal{L}(C)$ for an input prompt $C$. DIPPER or LLaMA-2-7B-Chat can be used to generate a paraphrase $\texttt{pp1}(S) = f_{strong}(S, C)$. This paraphrasing can be performed in recursion (see Figure 2). That is, $\texttt{pp2}(S) = f_{strong}(\texttt{pp1}(S), C)$ and so on.

While DIPPER is explicitly trained to be a paraphraser, LLaMA-2-7B-Chat is an instruction-tuned model for chat purposes. We design a system prompt (see Appendix B.2) with the LLaMA-2-7B-Chat model to use it as a paraphraser. In §2.3 and §2.4, we show that recursive paraphrasing is effective in evading the strong watermark and retrieval-based detectors when compared to a single round of paraphrasing. Using human and other automated evaluation techniques in §2.2, we show that our recursive paraphrasing method only degrades the text quality slightly most of the time.

## 2.2 Quality of the Paraphrases

In order to reliably study the quality of the recursive paraphrases we use in our experiments using DIPPER and LLaMA-2-7B-Chat, we perform human evaluations using MTurk and other automated techniques. The AI-text used in this study is generated using a watermarked OPT-13B model. Tables 1 and 2 provide a summary of the study. We investigate the content preservation and text quality or grammar of the recursive paraphrases with respect to the AI-generated texts (see Tables 7-10 in Appendix B.1 for more details). **In terms of content preservation with DIPPER, 70% of the paraphrases were rated high quality and 23% somewhat equivalent. In terms of text quality or grammar, 89% of the paraphrases were rated high quality.** On a Likert scale of 1 to 5, the DIPPER paraphrases that we use received an average rating of $4.14 \pm 0.58$ for text quality or grammar and $4.0 \pm 0.9$ for content preservation. **Similarly, 83% of the recursive paraphrases we obtain with LLaMA-2-7B-Chat were rated high quality.** See Appendix B.1 for more details on the human study.

For automated text quality evaluations, we use perplexity measures and a question-answering (QA) benchmark in Table 3. We measure the perplexity scores using OPT-13B. As shown in the table, the perplexity scores degrade from 5.5 to 8.7 and 10.5, respectively, for DIPPER and LLaMA-2-7B-Chat after 5 rounds of paraphrasing. We also use a QA benchmark SQuAD-v2 (Rajpurkar et al., 2016) to evaluate the effect of recursive paraphrasing. For this, we use two hundred data points from SQuAD-v2, which have context text length of at least 300 tokens. The length of context text passages we use in the study is $328 \pm 28$ tokens. Each data point consists of a context passage, a question, and an answer. We evaluate a QA model on the SquAD-v2 benchmark to observe that it achieves 97% accuracy in the QA task. For the QA model, we use the LLaMA-2-13B-Chat model with a carefully written system prompt (see Appendix B.2). To evaluate the quality

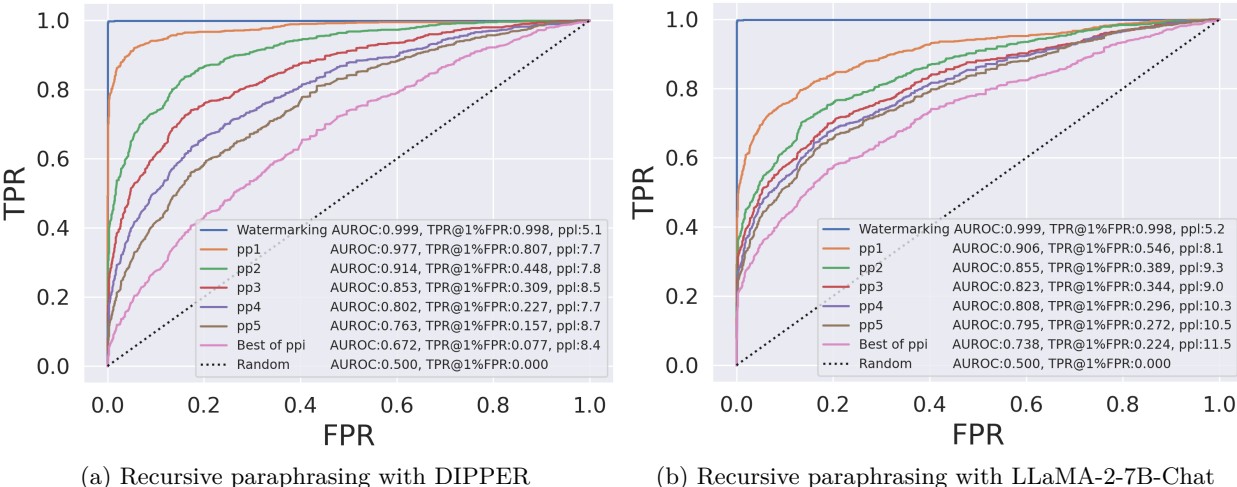

(a) Recursive paraphrasing with DIPPER

(b) Recursive paraphrasing with LLaMA-2-7B-Chat

Figure 3: ROC plots for soft watermarking with recursive paraphrasing attacks. AUROC, TPR@1%FPR, and perplexity scores measured using OPT-13B are given in the legend. The target LLM OPT-13B is used to generate watermarked output that are 300 tokens in length.

of paraphrases, we paraphrase the context passages recursively and use these to evaluate the QA accuracy with the QA model. If the QA model can answer the question correctly based on the paraphrased context, then the information is preserved in the paraphrase. As we observe, the QA performance with recursive paraphrasing is similar to that with the clean context passage. These results substantiate that AI text detectors can be effectively attacked using recursive paraphrasing with only a slight degradation in text quality.

We note that the amount of acceptable quality degradation can be application-specific. For example, an adversary might be okay with a higher quality drop when writing a social media post than when writing a fake news article. Our human studies rate our recursive paraphrases to have a score of either 5 or 4 over 70% of the time. Though this might be acceptable for some adversaries, others might not tolerate a score of 4 for their applications. Since a score of 4 denotes minor degradation, we presume that the adversaries could manually edit them for their attacks. Nevertheless, our paraphrases get a perfect score 35% of the time, indicating that it is still practical for adversaries to use it to perform their attacks successfully. However, we believe this tradeoff in text quality degradation and detection evasion would diminish as paraphrasers improve in the future.

## 2.3 Paraphrasing Attacks on Watermarked AI Text

In this section, we evaluate our recursive paraphrasing attacks on the soft watermarking scheme proposed in Kirchenbauer et al. (2023a). Soft watermarking encourages LLMs to output token $s^{(t)}$ at time-step $t$ that belongs to a "green list". The green list for $s^{(t)}$ is created using a private pseudo-random generator that is seeded with the prior token $s^{(t-1)}$. A watermarked output from the LLM is designed to have tokens that are majorly selected from the green list. Hence, a watermark detector with the pseudo-random generator checks the number of *green* tokens in a candidate passage to detect whether it is watermarked or not. Here, we target watermarked OPT-13B with 13B parameters in Figure 3 and watermarked OPT-1.3B in Figure 4 for our experiments. In Appendix A.2, we also evaluate our attacks on GPT-2 Medium (Radford et al., 2019) and other datasets – PubMedQA (Jin et al., 2019) and Kafkai (Kafkai, 2020).

**Dataset.** We perform our experiments on 2000 text passages that are around 300 tokens in length (1000 passages per human and AI text classes). We pick 1000 long news articles from the XSum "document" feature. For each article, the first ∼300 tokens are input to the target OPT-1.3B to generate 1000 watermarked AI text passages that are each ∼300 tokens in length. The second 300 tokens from the 1000 news articles in the dataset are treated as baseline human text. We note that our considered dataset has more and longer passages compared to the experiments in Kirchenbauer et al. (2023a).

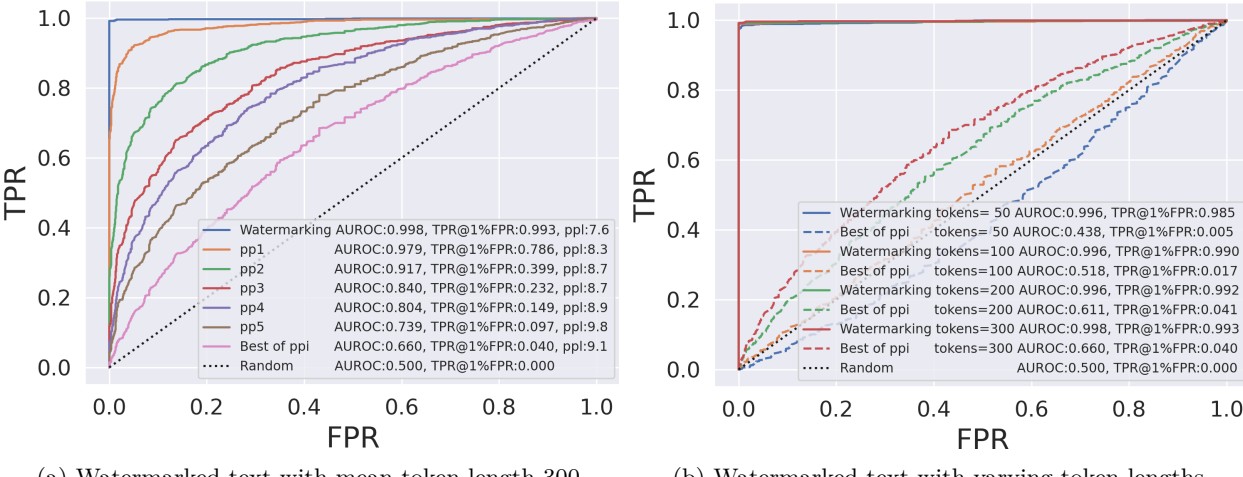

(a) Watermarked text with mean token length 300   (b) Watermarked text with varying token lengths

Figure 4: ROC plots for soft watermarking with recursive paraphrasing attacks. AUROC, TPR@1%FPR, and perplexity scores measured using OPT-13B are given in the legend. The target LLM is OPT-1.3B. (a) Even for 300 tokens long watermarked passages, recursive paraphrasing is effective. As paraphrasing rounds proceed, detection rates degrade significantly with a slight trade-off in text quality. (b) Attacking watermarked passages become easier as their length reduces.

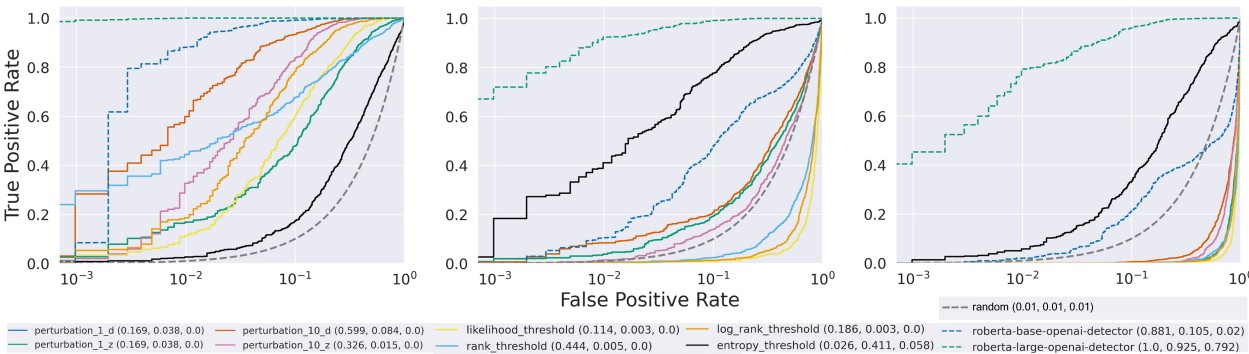

Figure 5: ROC curves for various trained and zero-shot detectors. **Left:** Without attack. **Middle:** After paraphrasing attack using T5-based paraphraser. The performance of zero-shot detectors drops significantly. **Right:** Here, we assume we can query the detector ten times for the paraphrasing attack. We generate ten paraphrasings for each passage and query multiple times to evade detection. Notice how all detectors have low TPR@1%FPR. In the plot legend – `perturbation` refers to the zero-shot methods in Mitchell et al. (2023); `threshold` refers to the zero-shot methods in Solaiman et al. (2019); Gehrmann et al. (2019); Ippolito et al. (2019); `roberta` refers to OpenAI's trained detectors (OpenAI, 2019). The TPR@1%FPR scores of different detectors before the attack, after the attack, and after the attack with multiple queries, respectively, are provided in the plot legend.

**Detection results after paraphrasing attack.** Weaker paraphrasing attacks discussed in Kirchenbauer et al. (2023a) are not effective in removing watermarks. They perform "span replacement" by replacing random tokens (in-place) using a language model. However, after a single round of paraphrasing (`pp1`) with a watermarked OPT-13B as the target LLM, TPR@1%FPR of watermark detector degrades from 99.8% to 80.7% and 54.6%, respectively, with DIPPER and LLaMA-2-7B-Chat paraphrasers. Though watermarking is a worthwhile endeavor to prevent AI plagiarism, with our stress test, we show that an adversary can find their way to evade detection via paraphrasing. As shown in Figures 3-4, the recursive paraphrase attack further degrades the detection rate of the detector to below 20% after 5 rounds of paraphrasing (`pp5`). Note that in all the settings `pp2` or 2 rounds of paraphrasing is sufficient to degrade TPR@1%FPR to below 50%. As shown in Figure 3, DIPPER shows a clearer and more consistent trend in improving attack performance over recursions

of paraphrasing in comparison to LLaMA-2. This is because DIPPER is trained explicitly to be a paraphraser with hyperparameters that can control the quality of paraphrasing. Therefore, we mainly employ DIPPER for our recursive paraphrase attacks. `Best of ppi` in the figure refers to the method where, for each passage, we select the paraphrase out of all the `ppi`'s that has the worst detector score. For `Best of ppi` with OPT-1.3B, the detection rate reduces drastically from 99.8% to 4.0% with a trade-off of 1.5 in the perplexity score (Figure 4a). `Best of ppi`, unlike the `ppi` attacks, assume black box query access to the detector. Figure 4b shows that the watermarking detector becomes weaker as the length of the watermarked text reduces. Note that for watermarked texts that are 50 or 100 tokens long, the detection performance after the recursive paraphrasing attack is similar to that of a random detector. As the plot indicates, watermarking could be more reliable for preventing AI plagiarism in tasks that require longer texts. However, this does not guarantee that watermarking will be a foolproof defense in such settings. This requires more investigation, and we leave this for future work. We provide examples of paraphrased text that we use for our attacks in Appendix B.3.

## 2.4 Paraphrasing Attacks on Non-Watermarked AI Text

Neural network-based trained detectors such as RoBERTa-Large-Detector from OpenAI (OpenAI, 2019) are trained or fine-tuned for binary classification with datasets containing human and AI-generated texts. Zero-shot classifiers leverage specific statistical properties of the source LLM outputs for their detection. Retrieval-based methods search for a candidate passage in a database that stores the LLM outputs. Here, we perform experiments on these non-watermarking detectors to show they are vulnerable to our paraphrasing attack.

**Trained and Zero-shot detectors.** We use a pre-trained GPT-2 Medium model (Radford et al., 2019) with 355M parameters as the target LLM to evaluate our attack on 1000 long passages from the XSum dataset (Narayan et al., 2018). We use the T5-based paraphrasing model (Damodaran, 2021) with 222M parameters to rephrase the 1000 output texts generated using the target GPT-2 Medium model.

Figure 5 shows the effectiveness of the paraphrasing attack over these detectors. **The AUROC scores of DetectGPT (Mitchell et al., 2023) drop from 96.5% (before the attack) to 59.8% (after the attack).** Note that AUROC of 50% corresponds to a random detector. The rest of the zero-shot detectors (Solaiman et al., 2019; Gehrmann et al., 2019; Ippolito et al., 2019) also perform poorly after our attack. Though the performance of the trained neural network-based detectors (OpenAI, 2019) is better than that of zero-shot detectors, they are also not reliable. For example, TPR@1%FPR of OpenAI's RoBERTa-Large-Detector drops from 100% to around 92% after our attack.

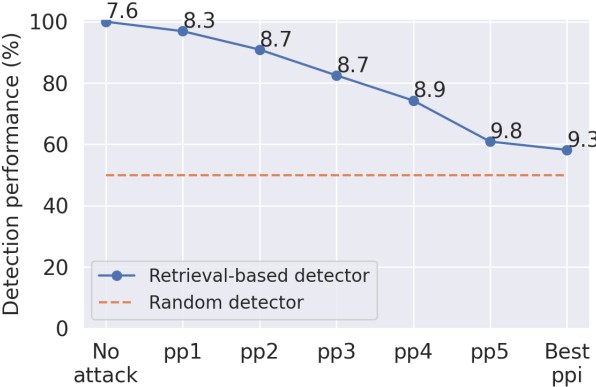

Figure 6: Recursive paraphrasing breaks the retrieval-based detector (Krishna et al., 2023) with only slight degradation in text quality. `ppi` refers to `i` recursion(s) of paraphrasing. Numbers next to markers denote the perplexity scores of the paraphraser output.

In another setting, we assume the attacker may have multiple access to the detector. That is, the attacker can query the detector with an input AI text passage, and the detector would reveal the detection score to the attacker. For this scenario, we generate ten different paraphrases for an input passage and query the detector for the detection scores. For each AI text passage, we then select the paraphrase with the worst detection score for evaluating the ROC curves. As shown in Figure 5, **with multiple queries to the detector, an adversary can paraphrase more efficiently to bring down TPR@1%FPR of the RoBERTa-Large-Detector from 100% to 80%.** In Appendix A.3, we show more experiments with more datasets and target LLMs.

As seen in the results, the detection of the entropy threshold detector improves with paraphrasing. LLMs are trained on human-written texts, and for this reason, they might have low entropy scores on human-written samples we use in our experiments due to memorization. Therefore, the entropy detector might have poor detection scores before the paraphrasing attack. However, after paraphrasing with a different AI model, the

entropy scores for these human-written samples might increase, improving the detection scores. Despite this, the entropy threshold detector has poor detection rates before and after the attack.

Though the performance of performance of trained detectors degrades after each round of paraphrasing, they seem to be more robust to paraphrase attacks than the other detectors we study. We hypothesize that this might be due to these detectors being trained on human-written samples we use for our study. For example, the MAGE dataset (Li et al., 2024) includes passages from the XSum dataset we use. Gameiro et al. (2024) argues that while trained detectors can generalize better to unseen LLMs, they may overfit to this training distribution of human text. They also show that some of these detectors fail to generalize to out-of-distribution human-written text. This is an aspect that we do not consider in our work, but would still make these detectors unreliable for real-world applications.

**Retrieval-based detectors.** Detector in Krishna et al. (2023) is designed to be robust against paraphrase attacks. However, we show that they can suffer from the recursive paraphrase attacks that we develop using DIPPER. We use 2000 passages (1000 generated by OPT-1.3B and 1000 human passages) from the XSum dataset. AI outputs are stored in the AI database by the detector. As shown in Figure 6, this detector detects almost all of the AI outputs even after a round of paraphrasing. However, **the detection accuracy drops below $\sim 60\%$ after five rounds of recursive paraphrasing.** As marked in the plot, the perplexity score of the paraphrased text only degrades by 1.7 at a detection accuracy of $\sim 60\%$. Moreover, retrieval-based detectors are concerning since they might lead to **serious privacy issues** from storing users' LLM conversations. In Appendix A.4, we show more experiments with more datasets and target LLMs.

## 3   Spoofing Attacks on Generative AI-text Models

An AI language detector without a low type-I error can cause harm as it might wrongly accuse a human of plagiarizing using an LLM. Moreover, an attacker (*adversarial human*) can generate a non-AI text to be detected as AI-generated. This is called the *spoofing attack*. An adversary can potentially launch spoofing attacks to produce derogatory texts to damage the reputation of the target LLM's developers. In this section, as a proof-of-concept, we show that current text detectors can be spoofed to detect texts composed by adversarial humans as AI-generated. More details on the spoofing experiments are presented in Appendix D.

**Soft watermarking.** As discussed in §2, soft watermarked LLMs (Kirchenbauer et al., 2023a) generate tokens from the "green list" that are determined by a pseudo-random generator seeded by the prefix token. Though the pseudo-random generator is private, an attacker can estimate the green lists by observing multiple token pairs in the watermarked texts from the target LLM. An adversarial human can then leverage the estimated green lists to compose texts by themselves that are detected to be watermarked. In our experiments, we estimate the green lists for 181 most commonly used words in the English vocabulary. We query the target watermarked OPT-1.3B model one million times to observe the token pair distributions within this smaller vocabulary subset we select. Note that this attack on a watermarked model only needs to be performed once to learn the watermarking pattern or the proxy green list to spoof it thereafter.

Based on the frequency of tokens that follow a prefix token in the observed generative outputs, we estimate green lists for each of the 181 common words. We build a tool that helps adversarial humans create watermarked sentences by providing them with the proxy green list that we learn with only access to a watermarked text corpora obtained from the target watermarked LLM. We observe that the **soft watermarking scheme can be spoofed to degrade its detection AUROC from 99.8% to 1.3%** (see Figure 7).

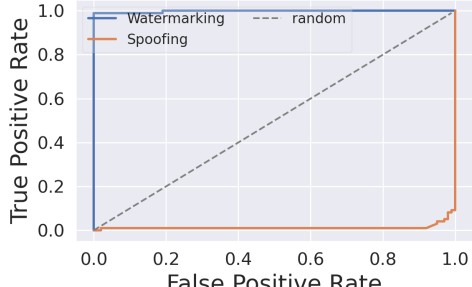

Figure 7: ROC curve of a soft watermarking-based detector (Kirchenbauer et al., 2023a) after our spoofing attack.

**Retrieval-based detectors.** Krishna et al. (2023) use a database to store LLM outputs to detect AI-text by retrieval. We find in our experiments (see Figure 13) that an **adversary can spoof this detector 100% of the time, even if the detector maintains a private database**. Suppose an adversary, say a teacher, has access to a human written document $S$, say a student's essay. The adversary

can prompt the target LLM to paraphrase $S$ to get $S'$. This results in the LLM, by design, storing its output $S'$ in its private database for detection purposes. Now, the detector would classify the original human text $S$ as AI-generated since a semantically similar copy $S'$ is present in its database. In this manner, a teacher can purposefully allege an innocent student to have plagiarised using the retrieval-based detector. Note that manipulating retrieval-based detectors is easier using this approach compared to watermarking techniques. This observation implies a practical tradeoff between type-I and type-II errors. When a detector is strengthened against type-II errors, it tends to result in a deterioration of its performance in terms of type-I errors.

**Zero-shot and neural network-based detectors.** In this setting, a malicious adversary could write a short text in a collaborative work, which may lead to the entire text being classified as AI-generated. To simulate this, we prepend a human-written text marked as AI-generated by the detector to all the other human-generated text for spoofing. In other words, from 200 long passages in the XSum dataset, we pick the human text with the worst detection score for each detector considered in §2.4. We then prepend this text to all the other human texts, ensuring that the length of the prepended text does not exceed the length of the original text. Our experiments show that the **AUROC of all these detectors drops after spoofing** (see plots in Appendix D). After this naïve spoofing attack, the TPR@1%FPR of most of these detectors drop significantly.

## 4 Hardness of Reliable AI Text Detection

In this section, we formally upper bound the AUROC of an arbitrary detector in terms of the TV between the distributions for $\mathcal{M}$ (e.g., AI text) and $\mathcal{H}$ (e.g., human text) over the set of all possible text sequences $\Omega$. We note that this result holds for any two arbitrary distributions $\mathcal{H}$ and $\mathcal{M}$. For example, $\mathcal{H}$ could be the text distribution for a person or group, while $\mathcal{M}$ could be the output text distribution of a general LLM or an LLM trained by an adversary to mimic the text of a particular set of people.

We use $\mathsf{TV}(\mathcal{M}, \mathcal{H})$ to denote the TV between these two distributions and model a detector as a function $D : \Omega \to \mathbb{R}$ that maps every sequence in $\Omega$ to a real number. Sequences are classified into AI-generated or human-generated by applying a threshold $\gamma$ on this number. By adjusting the parameter $\gamma$, we can tune the sensitivity of the detector to AI and human-generated texts to obtain an ROC curve.

**Theorem 1.** *The area under the ROC of any detector $D$ is bounded as*

$$\mathsf{AUROC}(D) \leq \frac{1}{2} + \mathsf{TV}(\mathcal{M}, \mathcal{H}) - \frac{\mathsf{TV}(\mathcal{M}, \mathcal{H})^2}{2}.$$

The proof is deferred to Appendix C.1. Figure 8 shows how the above bound grows as a function of the TV distance. This theorem states that as the TV distance between AI and human text distributions reduces, the AUROC of the best possible detector decreases. Based on our theory, an adversary can use advanced LLMs to mimic human text to reduce the TV distance between human and AI text distributions to evade text detection systems.

For a detector to have a good performance (say, AUROC > 0.9), the distributions of human and AI-generated texts must be very different from each other (TV > 0.5 based on the figure). As $\mathcal{M}$ gets more similar to $\mathcal{H}$ (say, TV < 0.2), the performance of even the best-possible detector becomes unreliable (AUROC < 0.7). For some applications, say AI-text plagiarism, reliable detection should have a low false positive rate (say, < 0.01) and a high true positive rate (say, > 0.9). Based on our theory, this cannot be achieved even when the overlap between the distributions is relatively low, say 11% (or TV = 0.9 − 0.01 = 0.89, based on equation 1 in Appendix C.1).

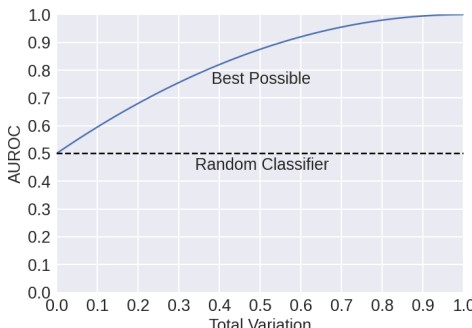

Figure 8: Comparing the performance, in terms of AUROC, of the best possible detector to that of the baseline performance corresponding to a random classifier.

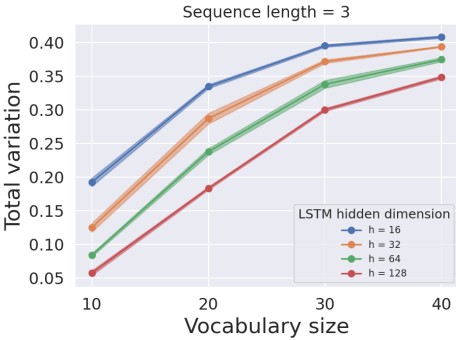

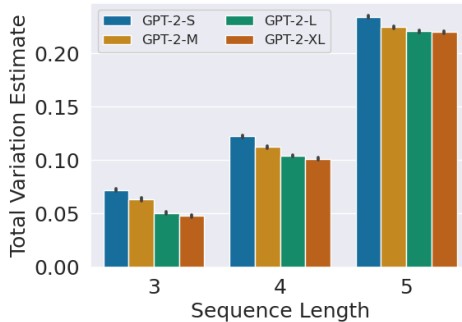

Figure 9: Increasing model size reduces the exact TV between the true synthetic data distribution and the learned distribution. Error bars report standard deviations after 5 independent trials.

Figure 10: Estimated TV distances of GPT-2 output datasets from the WebText dataset using meta-token sequences of varying lengths. TV decreases with model size for each length.

Note that, for a watermarked model, the above bound can be close to one as the TV between the watermarked distribution and human-generated distribution can be high. Corollary 1 in Appendix C.2 discusses how paraphrasing attacks can be effective in evading watermarks using Theorem 1. In Appendix C.3, we also present a tightness analysis of the bound in Theorem 1, where we show that for any distribution $\mathcal{H}$ there exists $\mathcal{M}$ and a detector $D$ for which the bound holds with equality. We also discuss general trade-offs between true positive and false positive rates of detection in Corollaries 2 and 3 in Appendix C.2. Theorem 2 in Appendix C.4 extends Theorem 1 to bound the AUROC of the best possible detector by a function of the TV distance between LLM outputs generated using pseudorandomness and human text distributions.

In studying the hardness of the detection problem, we consider the following assumption that for a given human-text distribution $\mathcal{H}$, more advanced LLMs mimicking $\mathcal{H}$ can lead to smaller TV. Thus, using Theorem 1, the detection problem becomes increasingly more difficult. This is the core argument of our hardness result on AI text detection. Although the underlying assumption seems to be intuitive given the capabilities of LLMs such as GPT-4 (OpenAI, 2023), a precise analysis of this assumption is quite difficult because estimating the true TV of the text distributions from a finite set of samples is extremely challenging. Nevertheless, we provide some empirical evidence supporting this assumption using two sets of experiments. In all the experiments, we consistently observe that the TV distance estimates between human and AI text distributions reduce as language models get more advanced, indicating the increasing difficulty associated with AI text detection.

**(i) Using synthetic text data.** We perform experiments on a toy synthetic text dataset where the exact TV distance can be calculated. We use the Markov assumption to generate the synthetic text data with sequence length 3 using a randomly generated token transition matrix for varying vocabulary sizes. We use single-layer LSTMs of different hidden unit sizes to train on a dataset of size 20,000 sampled from this synthetic data distribution using a default AdamW optimizer (Loshchilov & Hutter, 2017). We compute the learned token transition matrix for the LSTM output distribution using the softmax logit values of the trained model. Using transition matrices of both distributions, we compute the exact TV. Figure 9 shows that the exact TV distances between the learned and true synthetic distributions reduce as the LSTM model size increases.

**(ii) Using projection.** For discrete distributions, the TV distance can be computed as $1/2$ of the sum of the point-wise differences between their probability density functions (PDFs). While this is mathematically simple since texts can be considered as token sequences with bounded length, it is not practical to compute true TV distances directly through estimating PDFs due to the size of the sample space, which is approximately *the size of the token set* to the power of *sequence length*. To tackle this issue, we split the original token set into five roughly equal partitions and assign a meta-token to each partition. Given a sequence of tokens from the original set, we construct a new sequence by replacing each token with the corresponding meta-token. We estimate the PDFs of the sequences of meta-tokens created using texts from the WebText and GPT-2 output datasets. Since the set of meta-tokens is significantly smaller than the original token set, estimating PDFs becomes much more tractable. We then use these PDFs to estimate the total variation distances of the output

distributions of different GPT-2 models (GPT-2-Small, GPT-2-Medium, GPT-2-Large, and GPT-2-XL) from the WebText dataset. Figure 10 plots these TV estimates for different sequence lengths, averaged over 30 runs of the experiment. We observe that the TV distance consistently decreases with increasing model size for all sequence lengths.

These experiments provide empirical evidence that more advanced LLMs can lead to smaller TV distances. Thus, based on Theorem 1, reliable AI text detection would become increasingly difficult.

## Acknowledgments and Disclosure of Funding

This project was supported in part by NSF CAREER AWARD 1942230, ONR YIP award N00014-22-1-2271, NIST 60NANB20D134, Meta award 23010098, HR001119S0026 (GARD), Army Grant No. W911NF2120076, a capital one grant, and the NSF award CCF2212458 and an Amazon Research Award. Sadasivan is also supported by the Kulkarni Research Fellowship. The authors would like to thank Keivan Rezaei and Mehrdad Saberi for their insights on this work. The authors also acknowledge the use of OpenAI's ChatGPT to improve clarity and readability.

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

## A   Experiments with More Datasets and Models

In this section, we consider multiple datasets (XSum (Narayan et al., 2018), PubMedQA (Jin et al., 2019), and Kafkai (Pu et al., 2023)) and target LLMs (OPT-1.3B (Zhang et al., 2022) and GPT-2-Medium (Radford et al., 2019)) for analyzing our attacks.

**Datasets.** As discussed in the § 2.1, we use 2000 text passages (1000 passages each for human and AI-generated text classes) of ∼300 tokens in length from the XSum dataset for analyzing our attacks. For the rest of the datasets, we use 1000 text passages (500 passages each for human and AI-generated text classes) of ∼200 tokens in length. XSum contains long news articles in its "document" feature. To evaluate the

robustness of our attacks to distribution shifts, we include more datasets. We use PubMedQA, which is a medical text dataset. Kafkai dataset (Pu et al., 2023) contains real and fake articles (generated using privately fine-tuned OpenAI models) from 10 different domains, such as cybersecurity, SEO, and marketing. It is generated using Kafkai text generation service (Kafkai, 2020).

## A.1 Additional Experiments with Llama-2-13B

In this section, we provide evasion attack results with Llama-2-13B as the target model. We use a smaller Llama-2-7B model for recursive paraphrasing. In this section, we consider the XSum dataset for the experiments. As shown in Table 4, the detection performance (TPR@1%FPR) of detectors degrade with recursive paraphrasing.

| Detector | perturbation_1_d | perturbation_1_z | perturbation_10_d | perturbation_10_z | likelihood_threshold | rank_threshold | log_rank_threshold | entropy_threshold | MAGE | Longformer | roberta-base | roberta-large | Kuditipudi et al. |
|---|---|---|---|---|---|---|---|---|---|---|---|---|---|
| No attack | 0.32 | 0.32 | 0.612 | 0.048 | 0.992 | 0.148 | 0.98 | 0.0 | 0.457 | 0.772 | 0.672 | 0.648 | 0.988 |
| pp1 | 0.1 | 0.1 | 0.204 | 0.0 | 0.652 | 0.1 | 0.564 | 0.104 | 0.405 | 0.476 | 0.404 | 0.6 | 0.532 |
| pp2 | 0.142 | 0.142 | 0.124 | 0.02 | 0.536 | 0.076 | 0.444 | 0.116 | 0.4 | 0.454 | 0.316 | 0.556 | 0.356 |
| pp3 | 0.04 | 0.04 | 0.068 | 0.036 | 0.516 | 0.076 | 0.432 | 0.104 | 0.421 | 0.424 | 0.304 | 0.56 | 0.322 |
| pp4 | 0.04 | 0.04 | 0.052 | 0.02 | 0.492 | 0.08 | 0.412 | 0.12 | 0.427 | 0.436 | 0.296 | 0.504 | 0.292 |
| pp5 | 0.068 | 0.068 | 0.06 | 0.0 | 0.476 | 0.08 | 0.388 | 0.12 | 0.421 | 0.432 | 0.304 | 0.524 | 0.264 |

Table 4: Detector performance of various detectors where Llama-2-13B is the target model. The attacker uses a smaller Llama-2-7B model for recursive paraphrasing. The first four detectors with prefix "perturbation" are DetectGPT (Mitchell et al., 2023) variants. The next four detectors with suffix "threshold" are threshold-based zero-shot detectors. The next four detectors are trained detectors (Li et al., 2024; OpenAI, 2019). Kuditipudi et al. (2024) is a distortion-free watermarking technique.

## A.2 Watermark-based Detectors

In this section, we analyze the soft watermarking scheme in Kirchenbauer et al. (2023a). We use the powerful DIPPER paraphraser from Krishna et al. (2023) with 11B parameters for our recursive paraphrasing attack on the watermarking detector. On average, five rounds of our recursive paraphrase attack take around 36 seconds per text passage, 300 tokens in length. OPT-13B is used to measure the perplexity scores for all the settings. As shown in Table 1 and Appendix B.1, we perform a human study over the XSum dataset to evaluate the semantic drifts in our recursive paraphrasing framework. The MTurk human evaluation reveals that 70% of our recursive paraphrases maintain high-quality content preservation, and 89% of our recursive paraphrases have high-quality text or grammar.

Figure 11 shows the performance of the soft watermarking detector in multiple settings. In all the settings, the detection performance drops as rounds of recursive paraphrasing proceed with a slight degradation in perplexity scores. After two rounds of paraphrasing (pp2), the detection performance (TPR@1%FPR) in all the settings drops below 50%. `Best of ppi`, which selects the paraphrase with the worst detection score, significantly degrades the detection performance to below 10% in all the settings with degradation of 1.5, 0.5, 2.0, and 2.7 in perplexity measures.

## A.3 Zero-shot and Trained Detectors

In this section, we analyze the zero-shot and trained detectors in prior literature (Mitchell et al., 2023; Solaiman et al., 2019; Gehrmann et al., 2019; Ippolito et al., 2019; OpenAI, 2019). We use the T5-based paraphraser (Damodaran, 2021), Parrot, to paraphrase the AI-generated text and use OPT-13B to measure

the perplexity scores for all the settings. We perform our experiments on the XSum (Narayan et al., 2018), PubMedQA (Jin et al., 2019), and Kafkai (Kafkai, 2020) datasets with GPT-2-Medium and OPT-1.3B as the target generative models. In Figure 12 (ROC curves) and Tables 5 (TPR@1%FPR values) and 6 (AUROC scores), we present our results. 1d, 1z, 10d, and 10z in Tables 5 and 6 refer to different variants of the DetectGPT (Mitchell et al., 2023).

Figure 12 shows the performance of various zero-shot and trained detectors in multiple settings. The performance of these detectors drops significantly when the AI-generated text is paraphrased, and when given 5 queries to the detector, an adversary can fool most detectors effectively. Some detectors like OpenAI's RoBERTa-based models are more resilient on datasets like XSum, but are not reliable on other datasets like Kafkai. The perplexity scores of the GPT-2 generated text before any paraphrasing were 15.58 for XSum, 12.80 for PubMedQA, 19.11 for Kafkai, while the perplexity of OPT-1.3B generated text was 9.31. After paraphrasing, the perplexity scores are 20.06, 16.45, 20.01, and 13.96, respectively.

### A.4 Retrieval-based Detectors

In this section, we analyze the retrieval-based detector proposed in Krishna et al. (2023). We show that our recursive paraphrasing attack is effective in breaking their detector. We use the 11B parameter DIPPER paraphraser (Krishna et al., 2023) for our attack. OPT-13B is used to measure the perplexity scores in all the settings.

Figure 13 shows the performance of the retrieval-based detector in multiple settings. In all the settings, the detection accuracy drops as rounds of recursive paraphrasing proceed with only a slight degradation in perplexity scores. We observe that the detector works well after a single round of paraphrasing (`pp1`). However, after five rounds of paraphrasing, `Best of ppi` reduces the detector's accuracy to close to 50% with only a slight degradation in perplexity scores. We also find that we can easily spoof the retrieval-based detector as discussed in §3 to deteriorate the detector's performance to 0%. Note that retrieval-based detectors are concerning since they might lead to serious privacy issues from storing users' LLM conversations.

## B More Details on AI Paraphrasers

### B.1 Human Evaluation Study on Paraphrases

Apart from measuring the perplexity scores of the paraphrases using OPT-13B and performance on the SQUaD-v2 benchmark, we perform two human evaluation studies to investigate the quality of the paraphrases from DIPPER and LLaMA-2-7B-Chat we use for the paraphrasing attack. We pick 20 random watermarked passages and their corresponding five rounds of recursive paraphrases (`pp1` to `pp5`) for each of the paraphrasers for human evaluation. Each paraphrase is evaluated by 3 unique MTurk workers. We use the same setup as Krishna et al. (2023) for our human study. As shown in Figure 14, users are given a source text with some highlighted portion. The non-highlighted portion of the source text is input into the target OPT-13B model that generates watermarked text which is highlighted for the user's reference. DIPPER or LLaMA-2 paraphrases of the highlighted text are provided as the paraphrasing. The user is supposed to evaluate the quality of the paraphrases with respect to the highlighted watermarked text. They are supposed to rate it on a Likert scale of 1 to 5. See Tables 7,8 for the evaluation summary on content preservation of DIPPER paraphrasers based on the user study. Tables 9,10 show the summary of the evaluation of text quality/grammar of the paraphrases. For the content preservation study, we use the following Likert scale: 5 – preserves the meaning of the source but differs in words and/or structure. 4 – preserves most information in the source but differs in some minor factual details. 3 – reserves some information in the source but differs in certain significant ways. 2 – topically related to the source but most information in the source is not preserved. 1 – not topically related. For the text quality or grammar quality study, we use the following Likert scale: 5 – the paraphrase has excellent grammar/quality with respect to the highlighted source. 4 – the paraphrase is clear and correct with minor grammatical errors. 3 – the paraphrase has few grammatical errors, but remains clear and comparable to highlighted source text. 2 – the paraphrase has significant number of grammatical errors, but remains understandable. 1 – the paraphrase is inferior to the highlighted source text with a lot of grammatical errors, may be difficult to comprehend.

| | DetectGPT | | | | Threshold by | | | | RoBERTa | |
|---|---|---|---|---|---|---|---|---|---|---|
| | 1 d | 1 z | 10 d | 10 z | Likeli-hood | Rank | Log Rank | Entropy | Base | Large |
| **OPT-1.3B on XSum** | | | | | | | | | | |
| No attack | 0.079 | 0.079 | 0.083 | 0.125 | 0.237 | 0.382 | 0.288 | 0.017 | 0.694 | 0.956 |
| pp attack | 0.014 | 0.014 | 0.018 | 0.006 | 0.006 | 0.006 | 0.006 | 0.326 | 0.025 | 0.479 |
| 5 pp attack | 0.0 | 0.0 | 0.005 | 0.0 | 0.004 | 0.001 | 0.002 | 0.202 | 0.003 | 0.244 |
| **GPT-2 on PubMedQA** | | | | | | | | | | |
| No attack | 0.05 | 0.05 | 0.598 | 0.481 | 0.085 | 0.379 | 0.144 | 0.029 | 0.748 | 0.902 |
| pp attack | 0.052 | 0.052 | 0.19 | 0.054 | 0.015 | 0.042 | 0.017 | 0.202 | 0.181 | 0.606 |
| 5 pp attack | 0.0 | 0.0 | 0.031 | 0.002 | 0.008 | 0.01 | 0.012 | 0.135 | 0.088 | 0.452 |
| **GPT-2 on Kafkai** | | | | | | | | | | |
| No attack | 0.077 | 0.077 | 0.669 | 0.625 | 0.088 | 0.352 | 0.085 | 0.0 | 0.048 | 0.006 |
| pp attack | 0.056 | 0.056 | 0.125 | 0.081 | 0.004 | 0.021 | 0.004 | 0.002 | 0.01 | 0.0 |
| 5 pp attack | 0.0 | 0.0 | 0.023 | 0.006 | 0.002 | 0.004 | 0.002 | 0.0 | 0.0 | 0.0 |
| **GPT-2 on XSum** | | | | | | | | | | |
| No attack | 0.169 | 0.169 | 0.599 | 0.326 | 0.114 | 0.444 | 0.186 | 0.026 | 0.881 | 1.0 |
| pp attack | 0.038 | 0.038 | 0.084 | 0.015 | 0.003 | 0.005 | 0.003 | 0.411 | 0.105 | 0.925 |
| 10 pp attack | 0.0 | 0.0 | 0.0 | 0.0 | 0.0 | 0.0 | 0.0 | 0.058 | 0.02 | 0.792 |

Table 5: TPR@1%FPR for trained and zero-shot detectors in different settings. For all attacks, we use the T5-based paraphraser. Here, "pp attack" refers to the paraphrasing attack where the AI output is paraphrased by the T5-based model. "i pp attack" refers to the setting where the attacker has black-box access to the detector. Here, the paraphraser generates "i" paraphrases for every passage, and the attacker selects the passage that has the worst detection score after "i" queries to the detector.

| | DetectGPT | | | | Threshold by | | | | RoBERTa | |
|---|---|---|---|---|---|---|---|---|---|---|
| | 1 d | 1 z | 10 d | 10 z | Likeli-hood | Rank | Log Rank | Entropy | Base | Large |
| **OPT-1.3B on XSum** | | | | | | | | | | |
| No attack | 0.769 | 0.769 | 0.9 | 0.859 | 0.918 | 0.844 | 0.943 | 0.482 | 0.974 | 0.998 |
| pp attack | 0.487 | 0.487 | 0.453 | 0.41 | 0.241 | 0.387 | 0.282 | 0.868 | 0.562 | 0.945 |
| 5 pp attack | 0.162 | 0.162 | 0.244 | 0.182 | 0.153 | 0.216 | 0.181 | 0.821 | 0.316 | 0.9 |
| **GPT-2 on PubMedQA** | | | | | | | | | | |
| No attack | 0.816 | 0.816 | 0.973 | 0.955 | 0.804 | 0.796 | 0.892 | 0.615 | 0.982 | 0.998 |
| pp attack | 0.671 | 0.671 | 0.796 | 0.743 | 0.4 | 0.497 | 0.494 | 0.798 | 0.823 | 0.98 |
| 5 pp attack | 0.33 | 0.33 | 0.601 | 0.541 | 0.327 | 0.314 | 0.409 | 0.752 | 0.712 | 0.967 |
| **GPT-2 on Kafkai** | | | | | | | | | | |
| No attack | 0.814 | 0.814 | 0.976 | 0.971 | 0.865 | 0.86 | 0.89 | 0.394 | 0.817 | 0.86 |
| pp attack | 0.661 | 0.661 | 0.757 | 0.742 | 0.497 | 0.719 | 0.515 | 0.651 | 0.486 | 0.629 |
| 5 pp attack | 0.353 | 0.353 | 0.532 | 0.513 | 0.412 | 0.627 | 0.426 | 0.576 | 0.358 | 0.53 |
| **GPT-2 on XSum** | | | | | | | | | | |
| No attack | 0.837 | 0.837 | 0.976 | 0.949 | 0.879 | 0.868 | 0.93 | 0.617 | 0.993 | 1.0 |
| pp attack | 0.566 | 0.566 | 0.587 | 0.524 | 0.171 | 0.277 | 0.23 | 0.916 | 0.726 | 0.995 |
| 10 pp attack | 0.115 | 0.115 | 0.202 | 0.177 | 0.075 | 0.104 | 0.108 | 0.744 | 0.464 | 0.983 |

Table 6: AUROC for trained and zero-shot detectors in different settings. Here, "pp attack" refers to the paraphrasing attack where the AI output is paraphrased by the T5-based model. "i pp attack" refers to the setting where the attacker has black-box access to the detector. Here, the paraphraser generates "i" paraphrases for every passage, and the attacker selects the passage that has the worst detection score after "i" queries to the detector.

Based on the evaluations, 70% and 83% of the paraphrases are rated high quality in terms of content preservation for DIPPER and LLaMA-2, respectively. 89% and 88% of the paraphrases are rated to have high-quality text/grammar for DIPPER and LLaMA-2, respectively. Our human study shows the tradeoff of our recursive paraphrase attack strength with text quality for the watermark-based detector.

### B.2 LLaMA-2 Chat Template

Below, we provide the chat template we use to employ LLaMA-2-7B-Chat as a paraphraser.

**System Prompt:** You are a paraphraser. You are given an input passage 'INPUT'. You should paraphrase 'INPUT' to print 'OUTPUT'. 'OUTPUT' shoud be diverse and different as much as possible from 'INPUT' and should not copy any part verbatim from 'INPUT'. 'OUTPUT' should preserve the meaning and content of 'INPUT' while maintaining text quality and grammar. 'OUTPUT' should not be much longer than 'INPUT'. You should print 'OUTPUT' and nothing else so that its easy for me to parse.
**User Prompt:** INPUT: [Add input passage here]

Below, we provide the chat template we use to employ LLaMA-2-13B-Chat as a question-answering model.

**System Prompt:** You are given a context 'C: [Add context passage here]' and a question 'Q: [Add question here]'. Let 'A' be the answer to question 'Q' solely based on context 'C'. You are given the true answer 'A1: [Add ground truth answer here]' for question 'Q'.
**User Prompt:** INPUT: Do answers 'A1' and 'A' match? You SHOULD only be printing either 'YES' or 'NO'.

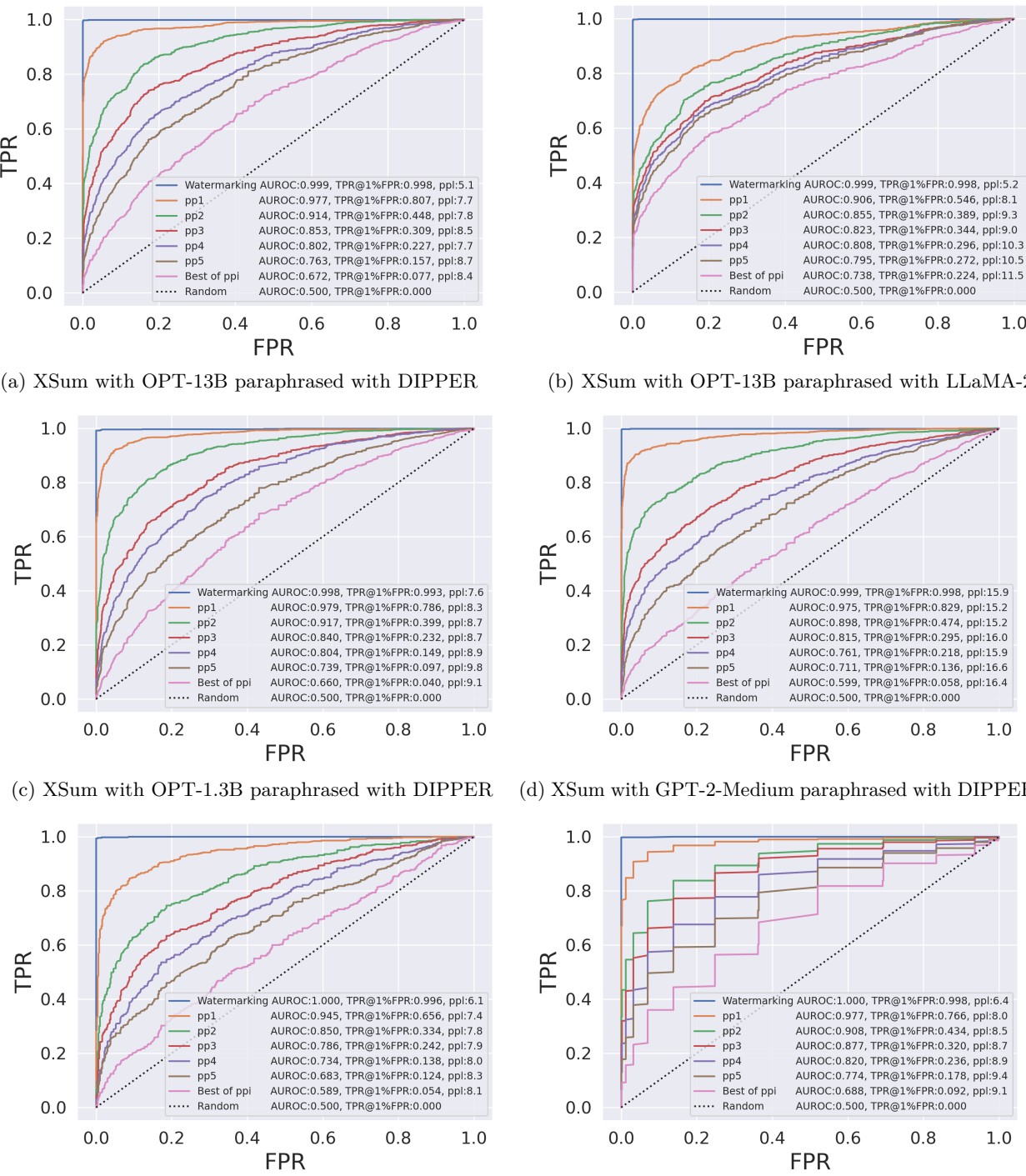

(a) XSum with OPT-13B paraphrased with DIPPER

(b) XSum with OPT-13B paraphrased with LLaMA-2

(c) XSum with OPT-1.3B paraphrased with DIPPER

(d) XSum with GPT-2-Medium paraphrased with DIPPER

(e) PubMedQA with OPT-1.3B paraphrased with DIPPER

(f) Kafkai with OPT-1.3B paraphrased with DIPPER

Figure 11: ROC plots for soft watermarking (Kirchenbauer et al., 2023a) with our recursive paraphrasing attacks. AUROC, TPR@1%FPR, and perplexity scores measured using OPT-13B are given in the legend. Detection performance on the XSum dataset using 3 different LLMs — OPT-13B, OPT-1.3B, and GPT-2-Medium — are evaluated in (a), (c), and (d), respectively. (b) show the performance of the detector with recursives paraphrases from LLaMA-2-7B-Chat. (e) and (f), respectively, show the performance of the detector on two datasets — PubMedQA and Kafkai — with distribution shifts using OPT-1.3B. In all the settings, we observe that the detection performance of the watermarking-based detector degrades but with a tradeoff in the text perplexity measures after the attack.

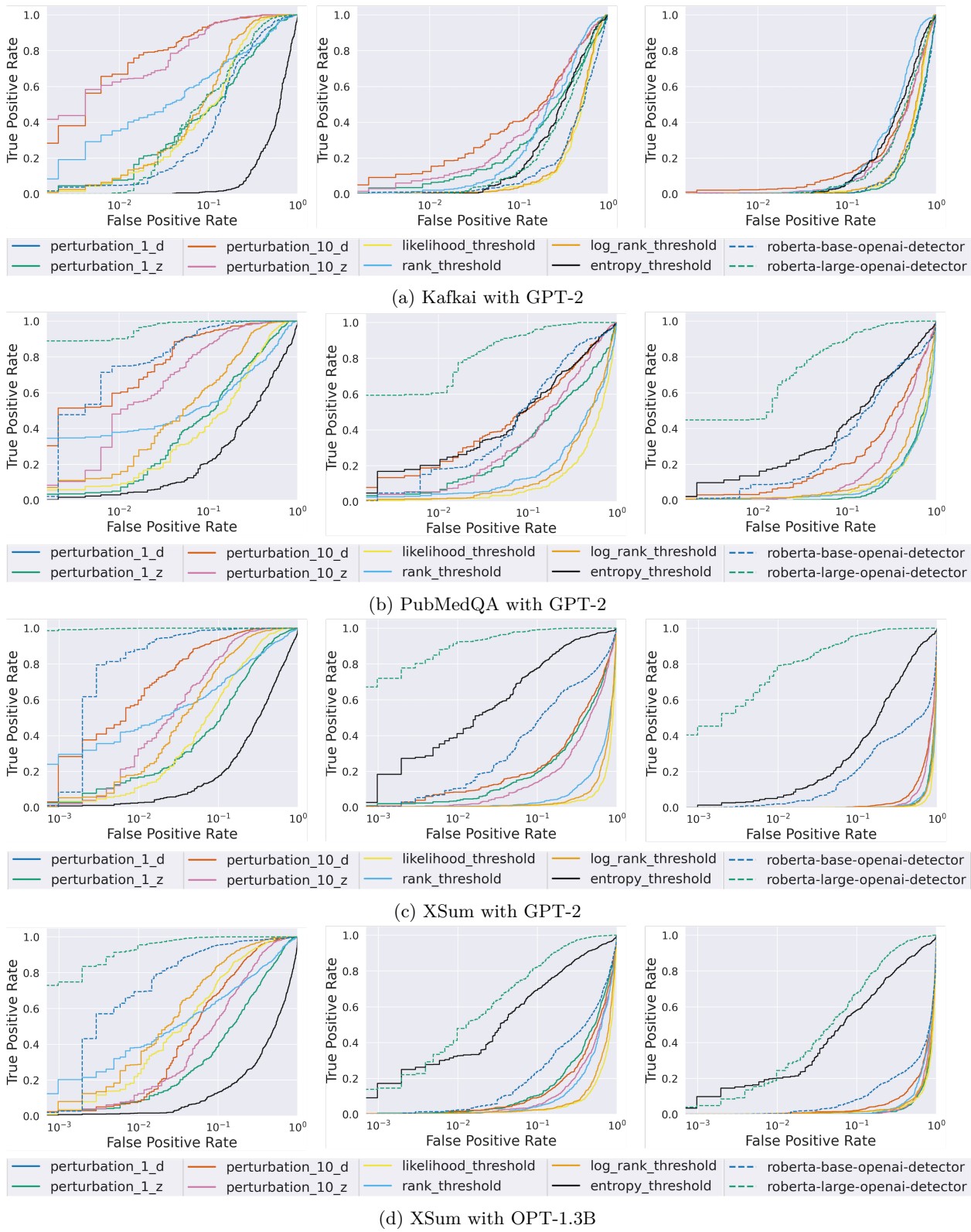

Figure 12: ROC curves for performance of various zero-shot and trained detectors for different models and datasets (**Left**) before attack, (**Middle**) after paraphrasing attack, (**Right**) applying paraphrasing attack with multiple queries to detector.

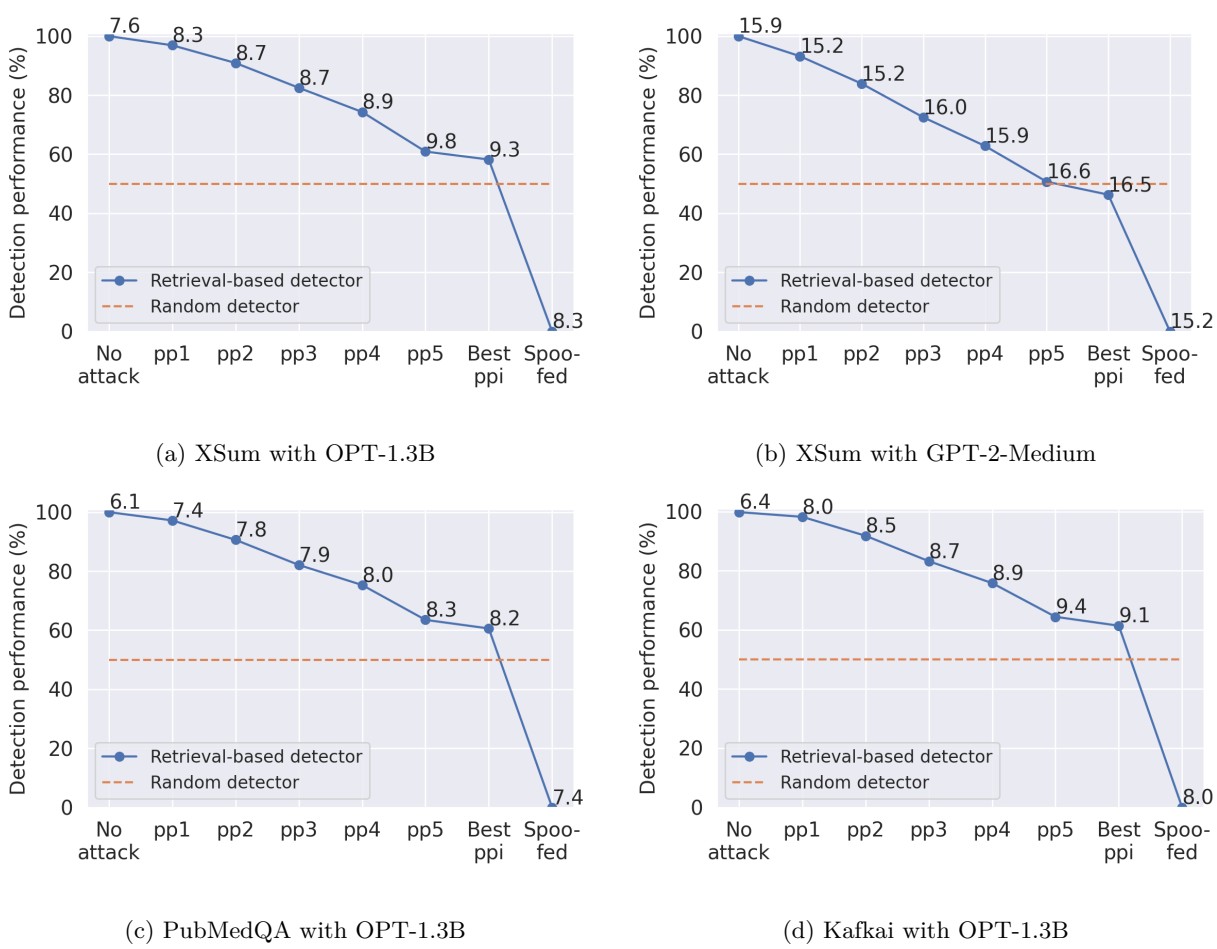

(a) XSum with OPT-1.3B

(b) XSum with GPT-2-Medium

(c) PubMedQA with OPT-1.3B

(d) Kafkai with OPT-1.3B

Figure 13: ROC plots for soft watermarking (Kirchenbauer et al., 2023a) with our recursive paraphrasing attacks. AUROC, TPR@1%FPR, and perplexity scores measured using OPT-13B are given in the legend. Detection performance on the XSum dataset using two different LLMs — OPT-1.3B and GPT-2-Medium — are evaluated in (a) and (b), respectively. (c) and (d), respectively, show the performance of the detector on two datasets — PubMedQA and Kafkai — with distribution shifts using OPT-1.3B. In all the settings, we observe that the detection performance of the watermarking-based detector reduces with a tradeoff in the perplexity measures of text after the attack.

View instructions

Read detailed instructions **carefully** before proceeding (click on "View Instructions" above).

## Given source text:

Forest Green, promoted from the National League, will host MK Dons in their first appearance in the competition, while FA Cup giant-killers Lincoln will be away to Rotherham. The 35 ties will be played in the week commencing Monday, 7 August. Hull City and Middlesbrough have been handed a bye into the second round, having finished above Sunderland in the Premier League last season. There was confusion after the draw, which was streamed live from Bangkok, where the competition's new sponsors, energy drink company Carabao, are based. A list of fixtures displayed on the stream showed Charlton drawn against two clubs, while AFC Wimbledon were also wrongly recorded as being at home to Swindon - the Dons were drawn at home to Brentford, and Swindon will be away to Norwich. And Forest Green were listed as being away to Wolves, who were in fact drawn at home to Yeovil. The live stream was also hampered by sound problems, with listeners on some clubs' websites unable to hear the draw. The draw was conducted by former Premier League referee Mark Clattenburg, who had been involved in the draw for the first match of the season. He is employed by Premier League broadcast partner Channel 5 as a television match official. The draw for the first round was conducted on live television at 10.30pm Thai time, with the match to be shown on Thailand's Channel 5 as part of the BBC's coverage of the tournament. The fourth round was drawn on live television at 10pm Thai time, with matches to be shown on Thailand's Channel 5 as part of the BBC's coverage of the tournament. The draw for the fifth round was conducted on live television at 10.25pm Thai time, with matches to be shown on Thailand's Channel 5 as part of the BBC's coverage of the tournament. The sixth round was drawn on live television at 10.15pm Thai time, with matches to be shown on Thailand' s Channel 5 as part of the BBC's coverage of the tournament. The seventh round was drawn on live television at 10.15pm Thai time, with matches to be shown on Thailand's Channel 5 as part of the BBC's coverage of the tournament. The draw for the eighth round was conducted on live television at 10.15pm Thai time, with matches to be shown on Thailand's Channel 5 as part of the BBC's coverage of the tournament. The draw for the ninth round was conducted on live television at 10.15pm Thai time, with matches to be

## Paraphrase of highlighted text:

Mark Clattenburg, a former Premier League referee, who officiated at the opening game of the season, was in charge of the draw. He is now employed by the FA's broadcast partner, Channel 5, as a television match official. The draw for the first round was made on live TV at 10.30pm local time, with Channel 5 in Thailand broadcasting the event as part of the BBC's coverage of the tournament. The fourth round was drawn live on Thai TV at 10pm local time, with matches to be screened on Channel 5 in Thailand as part of the BBC's coverage of the competition. The draw for the fifth round took place live on Thai TV at 10.25pm local time, with Channel 5 in Thailand to broadcast the ties as part of the BBC's coverage. The sixth round was drawn live on Thai TV at 10.15pm. Channel 5 in Thailand to show matches as part of BBC's coverage. The draw for the eighth round was made live on Thai TV at 10.15pm. Channel 5 in Thailand to show matches as part of BBC's coverage of the competition.

Which of the following best describes the quality of the paraphrase?

○ 5 - Approximately equivalent: the paraphrase preserves the meaning of the source but differs in words and/or structure
○ 4 - Nearly equivalent: the paraphrase preserves most information in the source but differs in some minor factual details
○ 3 - Somewhat equivalent: the paraphrase preserves some information in the source but differs in certain significant ways
○ 2 - Topically related: the paraphrase is topically related to the source but most information in the source is not preserved
○ 1 - Not topically related: the paraphrase is not topically related to the source and preserves no information

Write a short reasoning for your answer:

Write here...

Submit

Figure 14: MTurk user interface for human evaluation of paraphrases for content preservation.

| ppi | Average rating | Sum of 5 & 4 (%) | 5 - Approx. equivalent (%) | 4 - nearly equivalent (%) | 3 - Somewhat equivalent (%) | 2 - Topically related (%) | 1 - Topically unrelated (%) |
|---|---|---|---|---|---|---|---|
| i=1 | 4.0 ± 0.8 | 70.2 | 29.8 | 40.4 | 29.8 | 0.0 | 0.0 |
| i=2 | 4.1 ± 0.8 | 77.2 | 33.3 | 43.9 | 19.3 | 3.5 | 0.0 |
| i=3 | 3.9 ± 0.9 | 63.2 | 33.3 | 29.8 | 33.3 | 3.5 | 0.0 |
| i=4 | 4.2 ± 0.9 | 80.0 | 49.1 | 30.9 | 14.5 | 5.5 | 0.0 |
| i=5 | 3.7 ± 1.1 | 61.4 | 29.8 | 31.6 | 21.1 | 17.5 | 0.0 |
| **All ppi** | **4.0 ± 0.9** | **70.4** | **35.1** | **35.3** | **23.6** | **6.0** | **0.0** |

Table 7: MTurk human evaluation of recursive paraphrases with DIPPER for content preservation. `ppi` represents the $i^{th}$ round of recursive paraphrasing.

| ppi | Average rating | Sum of 5 & 4 (%) | 5 - Approx. equivalent (%) | 4 - nearly equivalent (%) | 3 - Somewhat equivalent (%) | 2 - Topically related (%) | 1 - Topically unrelated (%) |
|---|---|---|---|---|---|---|---|
| i=1 | 4.37 ± 0.63 | 91.67 | 45.0 | 46.67 | 8.3 | 0.0 | 0.0 |
| i=2 | 4.18 ± 0.67 | 85.0 | 33.33 | 51.67 | 15.0 | 0.0 | 0.0 |
| i=3 | 3.93 ± 0.71 | 80.0 | 21.67 | 58.33 | 16.67 | 3.3 | 0.0 |
| i=4 | 3.85 ± 0.78 | 78.33 | 21.67 | 56.67 | 16.67 | 5.0 | 0.0 |
| i=5 | 3.7 ± 1.1 | 80.0 | 18.33 | 61.67 | 11.67 | 8.33 | 0.0 |
| **All ppi** | **4.05 ± 0.2** | **83.0** | **28.0** | **55.0** | **13.67** | **3.33** | **0.0** |

Table 8: MTurk kuman evaluation of recursive paraphrases with LLaMA-2-7B-Chat for content preservation. `ppi` represents the $i^{th}$ round of recursive paraphrasing.

| ppi | Average rating | Sum of 5 & 4 (%) | 5 - Excellent (%) | 4 - Good (%) | 3 - Fair (%) | 2 - Adequate (%) | 1 - Poor (%) |
|---|---|---|---|---|---|---|---|
| i=1 | 4.28 ± 0.67 | 87.72 | 40.35 | 47.37 | 12.28 | 0.00 | 0.00 |
| i=2 | 4.12 ± 0.50 | 92.98 | 19.30 | 73.68 | 7.02 | 0.00 | 0.00 |
| i=3 | 4.12 ± 0.53 | 91.23 | 21.05 | 70.18 | 8.77 | 0.00 | 0.00 |
| i=4 | 4.11 ± 0.64 | 84.21 | 26.32 | 57.89 | 15.79 | 0.00 | 0.00 |
| i=5 | 4.07 ± 0.53 | 89.47 | 17.54 | 71.93 | 10.53 | 0.00 | 0.00 |
| **All ppi** | **4.14 ± 0.58** | **89.12** | **24.91** | **64.21** | **10.88** | **0.0** | **0.0** |

Table 9: Human evaluation of recursive paraphrases using MTurk for text quality/grammar. `ppi` represents the $i^{th}$ round of recursive paraphrasing.

| ppi | Average rating | Sum of 5 & 4 (%) | 5 - Excellent (%) | 4 - Good (%) | 3 - Fair (%) | 2 - Adequate (%) | 1 - Poor (%) |
|---|---|---|---|---|---|---|---|
| i=1 | 4.62 ± 0.55 | 96.67 | 65.0 | 31.67 | 3.33 | 0.00 | 0.00 |
| i=2 | 4.28 ± 0.73 | 83.33 | 45.0 | 38.33 | 16.67 | 0.00 | 0.00 |
| i=3 | 4.26 ± 0.65 | 88.33 | 43.33 | 45.0 | 11.67 | 0.00 | 0.00 |
| i=4 | 4.22 ± 0.64 | 88.33 | 38.33 | 50.0 | 11.67 | 0.00 | 0.00 |
| i=5 | 4.17 ± 0.74 | 83.33 | 40.0 | 43.33 | 15.0 | 1.67 | 0.00 |
| **All ppi** | **4.32 ± 0.35** | **88.0** | **46.33** | **41.67** | **11.67** | **0.33** | **0.0** |

Table 10: Human evaluation of recursive paraphrases using MTurk for text quality/grammar. `ppi` represents the $i^{th}$ round of recursive paraphrasing.

## B.3 Example Paraphrases

| Description | Text |
|---|---|
| Input | The draw was conducted by former Premier League referee Mark Clattenburg, who had been involved in the draw for the first match of the season. He is employed by Premier League broadcast partner Channel 5 as a television match official. The draw for the first round was conducted on live television at 10.30pm Thai time, with the match to be shown on Thailand's Channel 5 as part of the BBC's coverage of the tournament. The fourth round was drawn on live television at 10pm Thai time, with matches to be shown on Thailand's Channel 5 as part of the BBC's coverage of the tournament. The draw for the fifth round was conducted on live television at 10.25pm Thai time, with matches to be shown on Thailand's Channel 5 as part of the BBC's coverage of the tournament. The sixth round was drawn on live television at 10.15pm Thai time, with matches to be shown on Thailand's Channel 5 as part of the BBC's coverage of the tournament. The seventh round was drawn on live television at 10.15pm Thai time, with matches to be shown on Thailand's Channel 5 as part of the BBC's coverage of the tournament. The draw for the eighth round was conducted on live television at 10.15pm Thai time, with matches to be shown on Thailand's Channel 5 as part of the BBC's coverage of the tournament. The draw for the ninth round was conducted on live television at 10.15pm Thai time, with matches to be |
| Paraphrase | Mark Clattenburg, a former Premier League referee, who officiated at the opening game of the season, was in charge of the draw. He is now employed by the FA's broadcast partner, Channel 5, as a television match official. The draw for the first round was made on live TV at 10.30pm local time, with Channel 5 in Thailand broadcasting the event as part of the BBC's coverage of the tournament. The fourth round was drawn live on Thai TV at 10pm local time, with matches to be screened on Channel 5 in Thailand as part of the BBC's coverage of the competition. The draw for the fifth round took place live on Thai TV at 10.25pm local time, with Channel 5 in Thailand to broadcast the ties as part of the BBC's coverage. The sixth round was drawn live on Thai TV at 10.15pm. Channel 5 in Thailand to show matches as part of BBC's coverage. The draw for the eighth round was made live on Thai TV at 10.15pm. Channel 5 in Thailand to show matches as part of BBC's coverage of the competition. |
| Input | "We had to cancel his operation and put off on him, he was supposed to have it on Christmas Eve." He said Josh and his family had been in touch with some people from the US and Australia who are in Belfast right now. He is one of the first people from the US to come to Belfast to meet and speak with Josh, and the rest of the family will visit from the US in the next few days. "Josh has become a worldwide man's man, and it has been incredible to see how everybody has taken to him. He's an ambassador for Irish people and for Ireland, and the people are really behind him. He's a great ambassador for Ireland and for his country and it has been really good." Josh was taken to Queen's University Hospital for surgery and they discovered and cut down his appendix, which had been causing him pain for more than a year. The surgery took place on January 11. He is still on painkillers, and the family are encouraging people to keep praying for his recovery. The hashtag #prayforjosh and the post and the message he messages are now being shared widely through social media by people from the US and Australia. The most recent tweets from his father's account are about his recovery in Belfast. He is also being accompanied by a "pray warriors" team and the hashtag #prayforjosh is now being used by other campaigners to urge people to send support to Josh. The church is also asking people to help |
| Paraphrase | "His operation was cancelled and put back - it was due to be on Christmas Eve." He said that the family and friends of Josh have been in touch with people from the United States and Australia who are currently in Belfast. He himself was one of the first to arrive in the city to meet and talk to the teenager, and more of the family will arrive in a few days. "Josh has become a 'global' man and it's been amazing to see how people have taken to him. He's a real Irishman and an ambassador for Ireland and it's been really good for him." He said that on January 11th, when he was taken to Queen's University Hospital in Belfast for the operation, a large primary cancer and five smaller growths were removed. He is still recovering and the family ask people to keep praying for his recovery. The post with the hashtag #prayforjosh and the messages it contains have now been shared all over social media by people in the US and Australia. The last few posts on his father's Twitter account are about his recovery in Belfast. He has also been surrounded by a team of prayer warriors and the campaign has now widened, with people using the tag #prayforjosh to urge others to show their support for him. The church is also calling on people to help in other ways. |
| Input | I've enjoyed my time with the team." Wales, who won 11 medals in total in Brazil, failed to make the podium at three consecutive Games from 1996 to 2008 before Doull stepped into the team and helped to end the long wait for a Welsh team pursuit gold. The Welsh team were in fine form on Saturday as they captured their 11th medal, equalling the record of 14 won in 1972. The Welsh team's run to gold this year was well-documented. Geraint Thomas, the 2008 and 2012 Olympic champion, set an Olympic record of 31 minutes 59.4 seconds to help Great Britain to gold. Born in Cardiff but living in Aberystwyth, Anthony Iwan Thomas was selected as one of the two Welsh track riders for Rio and earned a silver medal in the men's road race, which he won in August. "It's a dream come true to come back and win Olympic gold here," said Thomas. "To do it in this stadium, it's just surreal. I can't explain it. Let me get home and I'll be really grateful to everyone." He missed the 2012 Games through injury but returned to gold-winning form in 2016 when he won his third consecutive world championship in the men's road race, finishing second in Rio. "It's a real honour to be on the podium today," added the 35-year-old. "There's a lot of times when you think 'this is it' and you think |
| Paraphrase | I've loved every minute with the team. " Wales, who won 11 medals in Brazil, had not finished on the podium at three consecutive Olympic games from 1996 to 2008 before Doull helped end the country's long wait for a team pursuit gold medal. The nation's 11th medal on the last day equalled the record of 14 set in 1972. So many stories of success for the Welsh riders in Rio have already been written. Born in Cardiff but now based in the Ceredigion resort of Aberystwyth, the son of Geraint Thomas won silver in the men's road race. Thomas said: "It's just a dream come true to come back and win gold here and it's even more special to do it in this stadium. " If I get home I'll thank everybody. " The 38-year-old had to miss the 2012 Olympic games with injury but was back to winning ways in 2016 when he retained his world title and finished second in Rio. "There's so many times when you think, 'This is the day' and it never comes." |

Table 11: Examples of paraphrased passage from the XSum dataset. The paraphrasing is performed using DIPPER (Krishna et al., 2023).

| Description | Text |
|---|---|
| Input | The force has said it has no intention of changing policy or use of force, and will continue to use the weapons as part of its role policing the public. A spokesman for Scotland Yard said: "Officers in Scotland will continue to have a range of weapons on-hand, including tasers, pepper spray and a handgun and they will be used in accordance with Taser International guidelines. "We also have well established policies governing the use of these weapons and the use of force, and the use of force is judged to be proportionate when appropriate." The move was welcomed by a leading rights campaign group. John O'Donnell, the head of the High Court, said Tasers should have been compulsory in Scotland in 2015 following the death of a journalist in custody in Newcastle and the deaths of two more while in police custody. He said: "The use of these weapons by police in Scotland has been widely condemned and the High Court ruled they should be used to protect the public." A spokesman for the Department for Transport said: "No police force should decide for itself what constitutes necessary force and the use of these weapons by police in Scotland has been widely condemned. "The use of these weapons by officers must be subject to strict rules which safeguard the public and the officers' rights." The London police said they "do not comment on intelligence matters".? |
| Paraphrase | The Met said it had no plans to change its policy on the use of force and said the Tasers would be used "when it is appropriate and proportionate". A spokesman said: "Officers in Scotland will continue to carry a range of weapons including Tasers, pepper spray and handguns. "We have well-established policies on the use of these weapons and the use of force in general and we will continue to ensure these are followed." It was welcomed by the Scottish Human Rights Commission. The high court judge, Lord O'Donnell, said in February that the use of Tasers by police in Scotland should have been compulsory in the wake of the death of a journalist in police custody in Newcastle, and the subsequent deaths of two more people in police custody. He said: "The use of these weapons by police in Scotland has been widely condemned and the high court has ruled they must be used to protect the public." The Department for Transport said police should not be able to decide for themselves when it was appropriate to use force. "The use of these weapons by officers must be subject to strict rules which protect the public and officers' rights." The Met said it did not comment on intelligence matters. Would you like to book an appointment with the British Transport Police? |
| Input | "He gets on with me. I have no doubt about his motives." The move is also an enormous gamble but Koukash, a self-made millionaire who made his fortune as a real estate mogul in Dubai, is determined to make sure the gamble pays off. He is willing, he says, to let the man who guided Salford to seven Grand Final appearances head a great project. "He has an incredible track record of creating people and businesses," said Noble. "He has done it in Qatar, in Dubai, in America, here." He is also one of the game's most ruthless businessmen. Koukash, who moved to the UK from Sudan with his parents as a boy and has spent the past 30 years building his empire of clubs and businesses, has seen the game of rugby league decline dramatically over the past 10 years. He claims that the game has never been more popular than it is today. He is also a fervent supporter of the game and this makes his interest in rugby league even more compelling. Salford have been in dire straits and Koukash, who has ploughed much of his own money in the club, has promised to help them become one of the great clubs of the game. The club has all the right qualities and Koukash wants to make sure it happens. He has spent the past two weeks scouting for players to recruit and has already seen the arrival of two promising youngsters. The Reds need players to make them competitive |
| Paraphrase | He added: "I know him. I have no doubt about his intentions." It's a huge gamble for the self-made millionaire who has made his money in property in Dubai. He is happy to entrust the man who took Salford to seven Challenge Cup finals with his great plan. " He's got an incredible record of turning around people and businesses," Noble said. "He's done it in Qatar, in Dubai, in America, here." He's also one of the most ruthless businessmen in the game. Having arrived in Britain as a boy from Sudan with his parents, he's spent the past 30 years building an empire of clubs and businesses. He's also seen the game decline in popularity over the past decade, but insists it's now more popular than ever. He is a huge supporter of the game and that's why he's interested in the sport. Salford have been in crisis and he's promised to help them become one of the great clubs. The club has all the attributes and Koukash is determined to make it happen. He has spent the past fortnight looking at players and has already recruited two. He has to make the Reds a more competitive side and has already brought in a couple of players who have impressed. |
| Input | She said he would not stop attacking and asked for help. She said she wanted "peace" but "not death". Henderson-McCarroll, of St Nicholas Drive, Newry, admitted manslaughter while in charge of a dangerous drug. She said her actions were a "blip in my mind" as a result of a "bad decision" to take drugs. Justice Treacy said he would not impose a custodial sentence on Henderson-McCarroll, but instead sentenced her to three years' imprisonment. The judge said he would not impose the maximum sentence for manslaughter given the circumstances, but felt he would not impose the minimum of two and a half years. He told Henderson-McCarroll: "He (Mr Girvan) would not be in his right mind if he would not have let his guard down. If there was one thing the jury should have heard - it was that your actions were a blip in my mind. You didn't intend to kill him. You were acting in self-defence. You poked him and your actions were a blip and a bit of a lapse in judgment." The judge said the maximum sentence for manslaughter given Henderson's previous convictions would have amounted to between five and seven years. The judge said it was "not an uncommon crime" to kill someone in self-defence. He said sentencing Henderson was an "ugly case of drug-induced madness." He added: "He (Mr Girvan) must have suffered terribly." |
| Paraphrase | She said he had not stopped attacking her and she called for help. She said she wanted "peace" but not death. Henderson-McCarroll, of St Nicholas Drive in Newry, admitted manslaughter while under the influence of a sedative. She said her actions were the result of a "mistake" after taking drugs. Mr Justice Treacy said he would not grant a suspended sentence, but instead would sentence her to three years in prison. He said he would not impose the maximum sentence for manslaughter, in the circumstances, but did not feel he should impose the minimum term of two and a half years. He said to the defendant: "You could not in your right mind have left your guard down, you did not intend to kill him. If there was one thing the jury ought to have heard, it was that your actions were a momentary lapse of reason. You acted in self defence, you poked him with a knife, your actions were momentary and a lapse of reason." The judge said that given the defendant's previous record the maximum sentence for manslaughter, with a minimum term of a year, would have been five to seven years. He said it was not an uncommon crime for someone to kill in self defence. " But in this case, it was an ugly case of drug induced madness. " He added: "Mr Girvan must have suffered horribly." |

Table 12: Examples of paraphrased passage from the XSum dataset. The paraphrasing is performed using DIPPER (Krishna et al., 2023).

| Description | Text |
|---|---|
| Input | In the year 2014-2015, Prison Link Cymru made 1,099 referrals, and said some offenders waited up to a year before finding suitable accommodation. Staff from the charitable organization say that the investment in housing would be cheaper than having to re-lock homeless offenders. The Welsh government said that more people than ever are getting help to deal with their housing problems. Changes to the Welsh Housing Act in 2015 removed priority rights for ex-offenders. Prison Link Cymru, which helps people to find accommodation after they have been released, said that women generally had a good time, as they had to deal with children and other problems. |
| pp1 | In the year 2014-2015, Prison Link Cymru made 1,099 referrals, and said some offenders waited up to a year before finding suitable accommodation. Staff from the charitable organization say that the investment in housing would be cheaper than having to re-lock homeless offenders. The Welsh government said that more people than ever are getting help to deal with their housing problems. Changes to the Welsh Housing Act in 2015 removed priority rights for ex-offenders. Prison Link Cymru, which helps people to find accommodation after they have been released, said that women generally had a good time, as they had to deal with children and other problems. |
| pp2 | In 2015-16, Prison Link Cymru had 1,099 referrals and said some offenders were living rough for up to a year before finding suitable accommodation. Workers at the charity claim investment in housing would be cheaper than locking up homeless repeat offenders. The Welsh Government said more people than ever before were getting help to address housing problems. Prison Link Cymru, which helps people find accommodation after they have left prison, said things were generally good for women because issues such as children or domestic violence were often the main reasons they returned to prison. |
| pp3 | In fact, it was one of the main reasons why they came back to prison. In the year 2015-16, Prison Link Cymru made 1,099 referrals and said that offenders had lived for up to a year before finding suitable accommodation. The workers at the charitable organization say that the investment in housing would be cheaper than re-locking homeless offenders. The government of Wales says that more people than ever are being helped to deal with their housing problems. Prison Link Cymru, which helps people to find accommodation after being released, says that women are generally in a good position, as children and other problems are often the main reason why they are returned to prison. |
| pp4 | In the year to the end of March, Prison Link Cymru had 1,099 referrals and said offenders had been living rough for up to a year before finding suitable accommodation. Workers at the charity say investment in housing would be cheaper than re-imprisoning homeless repeat offenders. The Welsh Government says more people than ever before are getting help to address their housing problems. Prison Link Cymru, which helps people find accommodation after they have been released from prison, said generally things were good for women because issues such as children and other problems were often the main reasons they returned to prison. |
| pp5 | The government of Wales says that more people than ever before are being helped to deal with their housing problems. In the year 2015-16, Prison Link Cymru referred 1,099 people and said that homeless people had lived up to a year before finding suitable accommodation. The workers at the charitable organization say that the investment in housing would be cheaper than imprisoning homeless offenders again. Prison Link Cymru, which helps people find accommodation after being released, says that women are generally well, because children and other problems are often the main reason why they return to prison. |

Table 13: Example of a recursively paraphrased passage from the XSum dataset. The paraphrasing is performed using DIPPER (Krishna et al., 2023). `ppi` refers to the output after `i` rounds of recursive paraphrasing.

| Description | Text |
|---|---|
| Input | The final round of fixtures will be played in Belfast on Tuesday. Elsewhere, Ireland's U19 women face Wales on Saturday in their Euro U19 women qualifier. The Irish women are already assured of a third-placed finish and a place at next year's finals in Northern Ireland after wins over Bulgaria and Greece. If they win against Wales they will finish second and qualify for a play-off for the finals. The Irish U19 women face Wales in Aberystwyth on Saturday. The match kicks off at 1.15pm. The Irish U19 women are in excellent form having won their last four matches. The winning margins have been 4-0, 4-0, 3-0, 3-0 and 3-0. The Irish U19 women sit fourth in the qualifying group and could qualify for the finals if they beat Wales on Saturday. The Irish women are in impressive form having won their last four games and qualified for next year's finals. Saturday's game against Wales is one of the most important of the campaign as a win will ensure they finish second and qualify for a play-off for the finals. The Irish will come up against whoever comes top of the group - Denmark, Portugal, Germany and Sweden - with the second place finisher going into a play-off for a final spot. The Irish U19 women took down Bulgaria 3-0, won 3-0 away to Greece and hammered home five against a young Spanish team. |
| pp1 | On Tuesday, the final round of matches will take place in Belfast. Meanwhile, Ireland's U19 women's team faces Wales in their Euro U19 women's qualifier on Saturday in Aberystwyth. The Irish are already assured of a third-placed finish and a spot in next year's finals after their wins over Bulgaria and Greece. A victory against Wales would see them finish second and qualify for a playoff for the finals. The Irish U19 women have been in impressive form, winning their last four matches by margins of 4-0, 4-0, 3-0, 3-0, and 3-0. They sit fourth in the qualifying group and could qualify for the finals if they beat Wales on Saturday. The Irish team will face the top finisher from the group - Denmark, Portugal, Germany, or Sweden - in a playoff for a final spot. The Irish U19 women have had a strong campaign so far, defeating Bulgaria 3-0, winning 3-0 away to Greece, and scoring five goals against a young Spanish team. |
| pp2 | On the eve of the final round of matches in Belfast, the Irish U19 women's team prepares to face Wales in their Euro U19 women's qualifier on Saturday in Aberystwyth. While Ireland has already secured a third-placed finish and a spot in next year's finals with their wins over Bulgaria and Greece, a victory against Wales would see them finish second and qualify for a playoff for the finals. The Irish U19 women have been in impressive form, winning their last four matches by margins of 4-0, 4-0, 3-0, 3-0, and 3-0. They currently sit fourth in the qualifying group and could potentially qualify for the finals if they beat Wales on Saturday. The team will face the top finisher from the group - Denmark, Portugal, Germany, or Sweden - in a playoff for a final spot. Ireland has had a strong campaign so far, defeating Bulgaria 3-0, winning 3-0 away to Greece, and scoring five goals against a young Spanish team. |
| pp3 | On the eve of the final round of matches in Belfast, the Northern Irish U19 women's team prepares to face Wales in their Euro U19 women's qualifier on Saturday in Aberystwyth. While Northern Ireland has already secured a third-placed finish and a spot in next year's finals with their wins over Spain and Scotland, a victory against Wales would see them finish second and qualify for a playoff for the finals. The Northern Irish U19 women have been in impressive form, winning their last four matches by margins of 4-0, 4-0, 3-0, and 3-0, and currently sit fourth in the qualifying group. They could potentially qualify for the finals if they beat Wales on Saturday, and face the top finisher from the group - Denmark, Portugal, Germany, or Sweden - in a playoff for a final spot. Northern Ireland has had a strong campaign so far, defeating Spain 2-1, winning 3-0 away to Scotland, and scoring five goals against a young Irish team. |
| pp4 | On the eve of the final round of matches in Northern Ireland, the local U19 women's team prepares to face Wales in their Euro U19 women's qualifier on Saturday in Cardiff. While Northern Ireland has already secured a third-placed finish and a spot in next year's finals with their recent wins, a victory against Wales would see them finish second and qualify for a playoff for the finals. The Northern Irish U19 women have been in impressive form, winning their last four matches by margins of 4-0, 4-0, 3-0, and 3-0, and currently sit fifth in the qualifying group. They could potentially qualify for the finals if they beat Wales on Saturday, and face the top finisher from the group - Denmark, Portugal, Germany, or Sweden - in a playoff for a final spot. Northern Ireland has had a strong campaign so far, defeating Spain 2-1, winning 3-0 away to Scotland, and scoring five goals against a young Irish team. |
| pp5 | On the eve of the crucial final match in Cardiff, the Northern Ireland U19 women's team prepares to face Wales in their Euro U19 women's qualifier on Saturday. With their recent string of victories, the Northern Irish have already secured a third-placed finish and a spot in next year's finals, but a win against Wales would see them finish second and qualify for a playoff for the finals. The team has been in impressive form, winning their last four matches by margins of 4-0, 4-0, 3-0, and 3-0, and currently sit fifth in the qualifying group. They could potentially qualify for the finals if they beat Wales on Saturday, and face the top finisher from the group - Denmark, Portugal, Germany, or Sweden - in a playoff for a final spot. Despite their strong campaign so far, the Northern Irish will face a tough challenge against Wales, who have also had a impressive run of form in the qualifiers. |

Table 14: Example of a recursively paraphrased passage from the XSum dataset. The paraphrasing is performed using LLaMA-2-7B-Chat (Touvron et al., 2023). `ppi` refers to the output after `i` rounds of recursive paraphrasing.

# C   Proofs and Corollaries

## C.1   Proof of Theorem 1

**Theorem 1.** *The area under the ROC of any detector $D$ is bounded as*

$$\mathsf{AUROC}(D) \leq \frac{1}{2} + \mathsf{TV}(\mathcal{M}, \mathcal{H}) - \frac{\mathsf{TV}(\mathcal{M}, \mathcal{H})^2}{2}.$$

*Proof.* The ROC is a plot between the true positive rate (TPR) and the false positive rate (FPR), which are defined as follows:

$$\mathsf{TPR}_\gamma = \mathbb{P}_{s \sim \mathcal{M}}[D(s) \geq \gamma]$$
$$\text{and } \mathsf{FPR}_\gamma = \mathbb{P}_{s \sim \mathcal{H}}[D(s) \geq \gamma],$$

where $\gamma$ is some classifier parameter. We can bound the difference between the $\mathsf{TPR}_\gamma$ and the $\mathsf{FPR}_\gamma$ by the total variation between $M$ and $H$:

$$|\mathsf{TPR}_\gamma - \mathsf{FPR}_\gamma| = |\mathbb{P}_{s \sim \mathcal{M}}[D(s) \geq \gamma] - \mathbb{P}_{s \sim \mathcal{H}}[D(s) \geq \gamma]| \leq \mathsf{TV}(\mathcal{M}, \mathcal{H}) \tag{1}$$
$$\mathsf{TPR}_\gamma \leq \mathsf{FPR}_\gamma + \mathsf{TV}(\mathcal{M}, \mathcal{H}). \tag{2}$$

Since the $\mathsf{TPR}_\gamma$ is also bounded by 1 we have:

$$\mathsf{TPR}_\gamma \leq \min(\mathsf{FPR}_\gamma + \mathsf{TV}(\mathcal{M}, \mathcal{H}), 1). \tag{3}$$

Denoting $\mathsf{FPR}_\gamma$, $\mathsf{TPR}_\gamma$, and $\mathsf{TV}(\mathcal{M}, \mathcal{H})$ with $x$, $y$, and $tv$ for brevity, we bound the AUROC as follows:

$$
\begin{aligned}
\mathsf{AUROC}(D) = \int_0^1 y \, dx &\leq \int_0^1 \min(x + tv, 1) dx \\
&= \int_0^{1-tv} (x + tv) dx + \int_{1-tv}^1 dx \\
&= \left| \frac{x^2}{2} + tvx \right|_0^{1-tv} + |x|_{1-tv}^1 \\
&= \frac{(1-tv)^2}{2} + tv(1 - tv) + tv \\
&= \frac{1}{2} + \frac{tv^2}{2} - tv + tv - tv^2 + tv \\
&= \frac{1}{2} + tv - \frac{tv^2}{2}.
\end{aligned}
$$

$\square$

## C.2   General Trade-offs For Detection

**Paraphrasing to Evade Detection.** Although our analysis considers general distributions, it can also be applied to specific scenarios, such as particular writing styles or sentence paraphrasing, by defining $\mathcal{M}$ and $\mathcal{H}$ appropriately. For example, $\mathcal{M}$ can be the outputs from an LLM trained to mimic a particular set of people, or $\mathcal{H}$ can be the text distribution of a specific person. Similarly, Corollary 1 shows that if a paraphraser's goal is to lower the TV between paraphrased AI text and human text, then detection gets harder.

Set $\mathcal{M} = \mathcal{R}_\mathcal{M}(s)$ and $\mathcal{H} = \mathcal{R}_\mathcal{H}(s)$ to be the distribution of sequences with similar meanings to $s$ produced by the paraphraser and humans, respectively.

**Corollary 1.** *The area under the ROC of the detector $D$ is bounded as*

$$\mathsf{AUROC}(D) \leq \frac{1}{2} + \mathsf{TV}(\mathcal{R}_\mathcal{M}(s), \mathcal{R}_\mathcal{H}(s)) - \frac{\mathsf{TV}(\mathcal{R}_\mathcal{M}(s), \mathcal{R}_\mathcal{H}(s))^2}{2}.$$

Another way to understand the limitations of AI-generated text detectors is directly through the characterization of the trade-offs between true positive rates and false positive rates. Adapting inequality 2, we have the following corollaries:

**Corollary 2.** *For any watermarking scheme $W$,*

$$\Pr_{s_w \sim \mathcal{R}_{\mathcal{M}}(s)}[s_w \text{ is watermarked using } W] \leq \Pr_{s_w \sim \mathcal{R}_{\mathcal{H}}(s)}[s_w \text{ is watermarked using } W]$$
$$+ \mathsf{TV}(\mathcal{R}_{\mathcal{M}}(s), \mathcal{R}_{\mathcal{H}}(s)),$$

*where $\mathcal{R}_{\mathcal{M}}(s)$ and $\mathcal{R}_{\mathcal{H}}(s)$ are the distributions of rephrased sequences for $s$ produced by the paraphrasing model and humans, respectively.*

Humans may have different writing styles. Corollary 2 indicates that if a rephrasing model resembles certain human text distribution $\mathcal{H}$ (i.e. $\mathsf{TV}(\mathcal{R}_{\mathcal{M}}(s), \mathcal{R}_{\mathcal{H}}(s))$ is small), then either certain people's writing will be detected falsely as watermarked (i.e. $\Pr_{s_w \sim \mathcal{R}_{\mathcal{H}}(s)}[s_w$ is watermarked using $W]$ is high) or the paraphrasing model can remove the watermark (i.e. $\Pr_{s_w \sim \mathcal{R}_{\mathcal{M}}(s)}[s_w$ is watermarked using $W]$ is low).

**Corollary 3.** *For any AI-text detector $D$,*

$$\Pr_{s \sim \mathcal{M}}[s \text{ is detected as AI-text by } D] \leq \Pr_{s \sim \mathcal{H}}[s \text{ is detected as AI-text by } D] + \mathsf{TV}(\mathcal{M}, \mathcal{H}),$$

*where $\mathcal{M}$ and $\mathcal{H}$ denote text distributions by the model and by humans, respectively.*

Corollary 3 indicates that if a model resembles certain human text distribution $\mathcal{H}$ (i.e. $\mathsf{TV}(\mathcal{M}, \mathcal{H})$ is small), then either certain people's writing will be detected falsely as AI-generated (i.e. $\Pr_{s \sim \mathcal{H}}[s$ is detected as AI-text by $D]$ is high) or the AI-generated text will not be detected reliably (i.e. $\Pr_{s \sim \mathcal{M}}[s$ is detected as AI-text by $D]$ is low).

A recent work (Chakraborty et al., 2023) shows a trade-off on the detection problem with respect to the availability of the number of data samples for detection. They show a TV upper bound for the detector's AUROC using an information theoretic approach. However, the underlying assumption of their result is that several *independent* samples are available to the detector from either human or text distribution, which might not be a practical assumption since sentences in a document are often correlated with each other. Also, a large number of data samples need not be available for pragmatic scenarios. For example, it may not be practical for a text detector to ask a student to write multiple essays for an assignment or to assume that a Twitter bot would publish longer tweets that are completely written by the AI without any human intervention.

### C.3 Tightness Analysis for Theorem 1

In this section, we show that the bound in Theorem 1 is tight. For a given distribution of human-generated text sequences $\mathcal{H}$, we construct an AI-text distribution $\mathcal{M}$ and a detector $D$ such that the bound holds with equality. Define sublevel sets of the probability density function of the distribution of human-generated text $\mathsf{pdf}_{\mathcal{H}}$ over the set of all sequences $\Omega$ as follows:

$$\Omega_{\mathcal{H}}(c) = \{s \in \Omega \mid \mathsf{pdf}_{\mathcal{H}}(s) \leq c\}$$

where $c \in \mathbb{R}$. Assume that, $\Omega_{\mathcal{H}}(0)$ is not empty. Now, consider a distribution $\mathcal{M}$, with density function $\mathsf{pdf}_{\mathcal{M}}$, which has the following properties:

1. The probability of a sequence drawn from $\mathcal{M}$ falling in $\Omega_{\mathcal{H}}(0)$ is $\mathsf{TV}(\mathcal{M}, \mathcal{H})$, i.e., $\mathbb{P}_{s \sim \mathcal{M}}[s \in \Omega_{\mathcal{H}}(0)] = \mathsf{TV}(\mathcal{M}, \mathcal{H})$.

2. $\mathsf{pdf}_{\mathcal{M}}(s) = \mathsf{pdf}_{\mathcal{H}}(s)$ for all $s \in \Omega(\tau) - \Omega(0)$ where $\tau > 0$ such that $\mathbb{P}_{s \sim \mathcal{H}}[s \in \Omega(\tau)] = 1 - \mathsf{TV}(\mathcal{M}, \mathcal{H})$.

3. $\mathsf{pdf}_{\mathcal{M}}(s) = 0$ for all $s \in \Omega - \Omega(\tau)$.

Define a hypothetical detector $D$ that maps each sequence in $\Omega$ to the negative of the probability density function of $\mathcal{H}$, i.e., $D(s) = -\mathsf{pdf}_{\mathcal{H}}(s)$. Using the definitions of $\mathsf{TPR}_{\gamma}$ and $\mathsf{FPR}_{\gamma}$, we have:

$$
\begin{aligned}
\mathsf{TPR}_{\gamma} &= \mathbb{P}_{s\sim\mathcal{M}}[D(s) \geq \gamma] \\
&= \mathbb{P}_{s\sim\mathcal{M}}[-\mathsf{pdf}_{\mathcal{H}}(s) \geq \gamma] \\
&= \mathbb{P}_{s\sim\mathcal{M}}[\mathsf{pdf}_{\mathcal{H}}(s) \leq -\gamma] \\
&= \mathbb{P}_{s\sim\mathcal{M}}[s \in \Omega_{\mathcal{H}}(-\gamma)]
\end{aligned}
$$

Similarly,

$$
\mathsf{FPR}_{\gamma} = \mathbb{P}_{s\sim\mathcal{H}}[s \in \Omega_{\mathcal{H}}(-\gamma)].
$$

For $\gamma \in [-\tau, 0]$,

$$
\begin{aligned}
\mathsf{TPR}_{\gamma} &= \mathbb{P}_{s\sim\mathcal{M}}[s \in \Omega_{\mathcal{H}}(-\gamma)] \\
&= \mathbb{P}_{s\sim\mathcal{M}}[s \in \Omega_{\mathcal{H}}(0)] + \mathbb{P}_{s\sim\mathcal{M}}[s \in \Omega_{\mathcal{H}}(-\gamma) - \Omega_{\mathcal{H}}(0)] \\
&= \mathsf{TV}(\mathcal{M}, \mathcal{H}) + \mathbb{P}_{s\sim\mathcal{M}}[s \in \Omega_{\mathcal{H}}(-\gamma) - \Omega_{\mathcal{H}}(0)] && \text{(using property 1)} \\
&= \mathsf{TV}(\mathcal{M}, \mathcal{H}) + \mathbb{P}_{s\sim\mathcal{H}}[s \in \Omega_{\mathcal{H}}(-\gamma) - \Omega_{\mathcal{H}}(0)] && \text{(using property 2)} \\
&= \mathsf{TV}(\mathcal{M}, \mathcal{H}) + \mathbb{P}_{s\sim\mathcal{H}}[s \in \Omega_{\mathcal{H}}(-\gamma)] - \mathbb{P}_{s\sim\mathcal{H}}[s \in \Omega_{\mathcal{H}}(0)] && (\Omega_{\mathcal{H}}(0) \subseteq \Omega_{\mathcal{H}}(-\gamma)) \\
&= \mathsf{TV}(\mathcal{M}, \mathcal{H}) + \mathsf{FPR}_{\gamma}. && (\mathbb{P}_{s\sim\mathcal{H}}[s \in \Omega_{\mathcal{H}}(0)] = 0)
\end{aligned}
$$

For $\gamma \in [-\infty, -\tau]$, $\mathsf{TPR}_{\gamma} = 1$, by property 3. Also, as $\gamma$ goes from 0 to $-\infty$, $\mathsf{FPR}_{\gamma}$ goes from 0 to 1. Therefore, $\mathsf{TPR}_{\gamma} = \min(\mathsf{FPR}_{\gamma} + \mathsf{TV}(\mathcal{M}, \mathcal{H}), 1)$ which is similar to Equation 3. Calculating the AUROC in a similar fashion as in the previous section, we get the following:

$$
\mathsf{AUROC}(D) = \frac{1}{2} + \mathsf{TV}(\mathcal{M}, \mathcal{H}) - \frac{\mathsf{TV}(\mathcal{M}, \mathcal{H})^2}{2}.
$$

### C.4 Pseudorandomness in LLMs

Most machine learning models, including LLMs, use pseudorandom number generators in one form or another to produce their outputs. For example, an LLM may use a pseudorandom number generator to sample the next token in the output sequence. In discussing our hardness result, Kirchenbauer et al. (2023b) in a more recent work argue that this pseudorandomness makes the AI-generated text distribution very different from the human-generated text distribution. This is because the pseudorandom AI-generated distribution is a collection of Dirac delta function distributions, and a human is exorbitantly unlikely to produce a sample corresponding to any of the delta functions. In our framework, this means that the TV between the human and pseudorandom AI-generated distributions is almost one, making the bound in Theorem 1 vacuous.

We argue that although the true TV between the human and pseudorandom AI-generated distributions is high and there exists (in theory) a detector function that can separate the distributions almost perfectly, this function may not be efficiently computable. Any polynomial-time computable detector can only achieve a negligible advantage from the use of pseudorandomness instead of true randomness. If we had knowledge of the seed used for the pseudorandom number generator, we would be able to predict the pseudorandom samples. However, an individual seeking to evade detection could simply randomize this seed making it computationally infeasible to predict the samples.

We modify the bound in Theorem 1 to include a negligible correction term $\epsilon$ to account for the use of pseudorandomness. We prove that the performance of a polynomial-time computable detector $D$ on a pseudorandom version of the AI-generated distribution $\widehat{\mathcal{M}}$ is bounded by the total variation for the truly random distribution $\mathcal{M}$ (resulting from the LLM using true randomness) as follows:

$$
\mathsf{AUROC}(D) \leq \frac{1}{2} + \mathsf{TV}(\mathcal{M}, \mathcal{H}) - \frac{\mathsf{TV}(\mathcal{M}, \mathcal{H})^2}{2} + \epsilon.
$$

The term $\epsilon$ represents the gap between the probabilities assigned by $\mathcal{M}$ and $\widehat{\mathcal{M}}$ to any polynomial-time computable $\{0,1\}$-function $f$, i.e.,

$$
\left| \mathbb{P}_{s\in\mathcal{M}}[f(s) = 1] - \mathbb{P}_{s\in\widehat{\mathcal{M}}}[f(s) = 1] \right| \leq \epsilon. \tag{4}
$$

This term is orders of magnitude smaller than any of the terms in the bound and can be safely ignored. For example, commonly used pseudorandom generators[1] can achieve an $\epsilon$ that is bounded by a negligible function $1/b^t$ of the number of bits $b$ used in the seed of the generator for a positive integer $t$[2] (Blum et al., 1982; Blum & Micali, 1984). From a computational point of view, the TV for the pseudorandom distribution is almost the same as the truly random AI-generated distribution. Thus, our framework provides a reasonable approximation for real-world LLMs, and the hardness result holds even in the presence of pseudorandomness.

**Computational Total Variation Distance.** Just as the total variation distance $\mathsf{TV}$ between two probability distributions is defined as the difference in probabilities assigned by the two distributions to any $\{0, 1\}$-function, we define a computational version of this distance $\mathsf{TV}_c$ for polynomial-time computable functions:

$$\mathsf{TV}_c(A, B) = \max_{f \in \mathcal{P}} \big| \mathbb{P}_{s \sim A}[f(s) = 1] - \mathbb{P}_{s \sim B}[f(s) = 1] \big|,$$

where $\mathcal{P}$ represents the set of polynomial-time computable $\{0, 1\}$-functions. $\mathcal{P}$ could also be defined as the set of all polynomial-size circuits which could be more appropriate for deep neural network-based detectors. The function $f$ could be thought of as the indicator function for the detection parameter being above a certain threshold, i.e., $D(s) \geq \gamma$ as in the proof of Theorem 1. The following lemma holds for the performance of a polynomial-time detector $D$:

**Lemma 1.** *The area under the ROC of any polynomial-time computable detector $D$ is bounded as*

$$\mathsf{AUROC}(D) \leq \frac{1}{2} + \mathsf{TV}_c(\widehat{\mathcal{M}}, \mathcal{H}) - \frac{\mathsf{TV}_c(\widehat{\mathcal{M}}, \mathcal{H})^2}{2}.$$

This lemma can be proved in the same way as Theorem 1 by replacing the truly random AI-generated distribution $\mathcal{M}$ with its pseudorandom version $\widehat{\mathcal{M}}$ and the true total variation $\mathsf{TV}$ with its computaional variant $\mathsf{TV}_c$.

Next, we relate the computational total variation $\mathsf{TV}_c$ between $\mathcal{H}$ and the pseudorandom distribution $\widehat{\mathcal{M}}$ with the total variation $\mathsf{TV}$ between $\mathcal{H}$ and the truly random distribution $\mathcal{M}$.

**Lemma 2.** *For human distribution $\mathcal{H}$, truly random AI-generated distribution $\mathcal{M}$ and its pseudorandom version $\widehat{\mathcal{M}}$,*

$$\mathsf{TV}_c(\widehat{\mathcal{M}}, \mathcal{H}) \leq \mathsf{TV}(\mathcal{M}, \mathcal{H}) + \epsilon.$$

*Proof.*

$$
\begin{aligned}
\mathsf{TV}_c(\widehat{\mathcal{M}}, \mathcal{H}) &= \max_{f \in \mathcal{P}} \big| \mathbb{P}_{s \sim \mathcal{H}}[f(s) = 1] - \mathbb{P}_{s \sim \widehat{\mathcal{M}}}[f(s) = 1] \big| && \text{(from definition of } \mathsf{TV}_c) \\
&= \max_{f \in \mathcal{P}} \big| \mathbb{P}_{s \sim \mathcal{H}}[f(s) = 1] - \mathbb{P}_{s \sim \mathcal{M}}[f(s) = 1] \\
&\quad + \mathbb{P}_{s \sim \mathcal{M}}[f(s) = 1] - \mathbb{P}_{s \sim \widehat{\mathcal{M}}}[f(s) = 1] \big| && (+/\text{-ing } \mathbb{P}_{s \sim \mathcal{M}}[f(s) = 1]) \\
&\leq \max_{f \in \mathcal{P}} \big| \mathbb{P}_{s \sim \mathcal{H}}[f(s) = 1] - \mathbb{P}_{s \sim \mathcal{M}}[f(s) = 1] \big| \\
&\quad + \big| \mathbb{P}_{s \sim \mathcal{M}}[f(s) = 1] - \mathbb{P}_{s \sim \widehat{\mathcal{M}}}[f(s) = 1] \big| && \text{(using } |a + b| \leq |a| + |b|) \\
&\leq \mathsf{TV}(\mathcal{M}, \mathcal{H}) + \epsilon. && \text{(from definition of } \mathsf{TV} \text{ and bound 4)}
\end{aligned}
$$

$\square$

We now use this to prove the modified version of our computational hardness result.

**Theorem 2 (Computational Hardness Result).** *The AUROC of any polynomial-time computable detector $D$ for $\mathcal{H}$ and the pseudorandom distribution $\widehat{\mathcal{M}}$ is bounded using the $\mathsf{TV}$ for the truly random distribution $\mathcal{M}$ as*

$$\mathsf{AUROC}(D) \leq \frac{1}{2} + \mathsf{TV}(\mathcal{M}, \mathcal{H}) - \frac{\mathsf{TV}(\mathcal{M}, \mathcal{H})^2}{2} + \epsilon.$$

---

[1]Cryptographic PRNGs:
https://en.wikipedia.org/wiki/Pseudorandom_number_generator
[2]Negligible function: https://en.wikipedia.org/wiki/Negligible_function

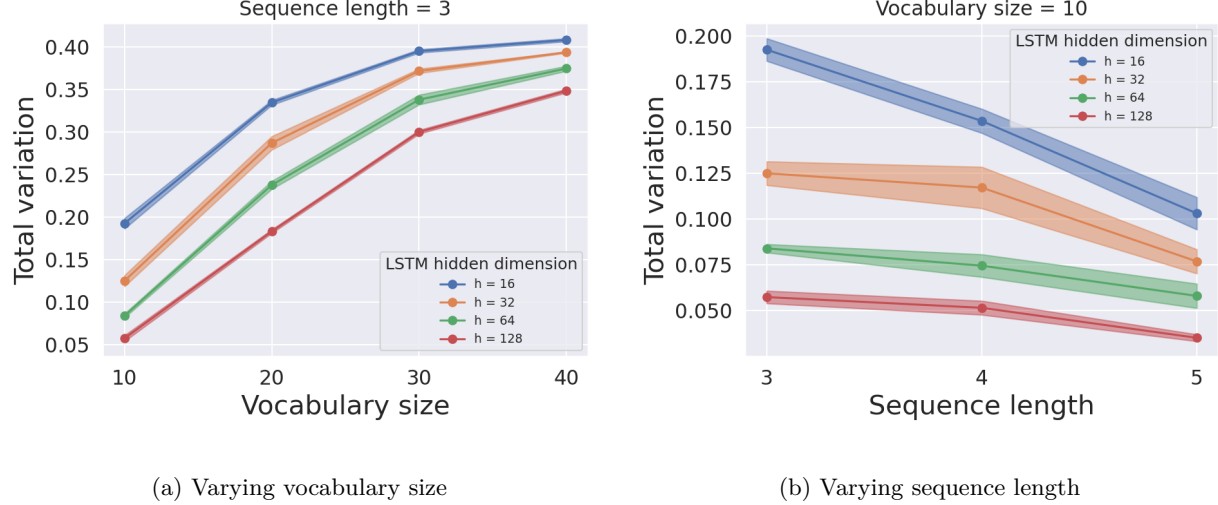

(a) Varying vocabulary size

(b) Varying sequence length

Figure 15: TV distances between synthetic toy data distributions and LSTM model generation distributions. TV distances are computed for multiple settings, varying the vocabulary size and sequence length of the training dataset and varying the size of the LSTM network used for training.

*Proof.*

$$\mathsf{AUROC}(D) \leq \frac{1}{2} + \mathsf{TV}_c(\widehat{\mathcal{M}}, \mathcal{H}) - \frac{\mathsf{TV}_c(\widehat{\mathcal{M}}, \mathcal{H})^2}{2} \qquad \text{(from Lemma 1)}$$

$$\leq \frac{1}{2} + \mathsf{TV}(\mathcal{M}, \mathcal{H}) + \epsilon - \frac{(\mathsf{TV}(\mathcal{M}, \mathcal{H}) + \epsilon)^2}{2}$$

$$\text{(from Lemma 2 and since } \tfrac{1}{2} + x - \tfrac{x^2}{2} \text{ is increasing in } [0, 1])$$

$$= \frac{1}{2} + \mathsf{TV}(\mathcal{M}, \mathcal{H}) + \epsilon - \frac{\mathsf{TV}(\mathcal{M}, \mathcal{H})^2 + \epsilon^2 + 2\epsilon\mathsf{TV}(\mathcal{M}, \mathcal{H})}{2}$$

$$\leq \frac{1}{2} + \mathsf{TV}(\mathcal{M}, \mathcal{H}) - \frac{\mathsf{TV}(\mathcal{M}, \mathcal{H})^2}{2} + \epsilon. \qquad \text{(dropping negative terms containing } \epsilon)$$

$\square$

### C.5 Estimating TV Distance

In §4, we show experiments supporting the assumption that more advanced LLMs lead to smaller TV distance between human and machine text distributions. We present two experimental settings — (i) Using synthetic text data and (ii) Using projection. In Figure 15, we show the TV distances computed with varying vocabulary sizes and sequence lengths. In all the experiments, we consistently find that the TV distances reduce as the network size increases.

## D   More Details on Spoofing

### D.1   Soft Watermark Detector

In Kirchenbauer et al. (2023a), they watermark LLM outputs by asserting the model to output tokens with some specific pattern that can be easily detected with meager error rates. Soft watermarked texts are majorly composed of *green list* tokens. If an adversary can learn the green lists for the soft watermarking scheme, they can use this information to generate human-written texts that are detected to be watermarked. Our experiments show that the soft watermarking scheme can be spoofed efficiently. Though the soft watermarking detector can detect the presence of a watermark very accurately, it cannot be certain if this pattern is actually

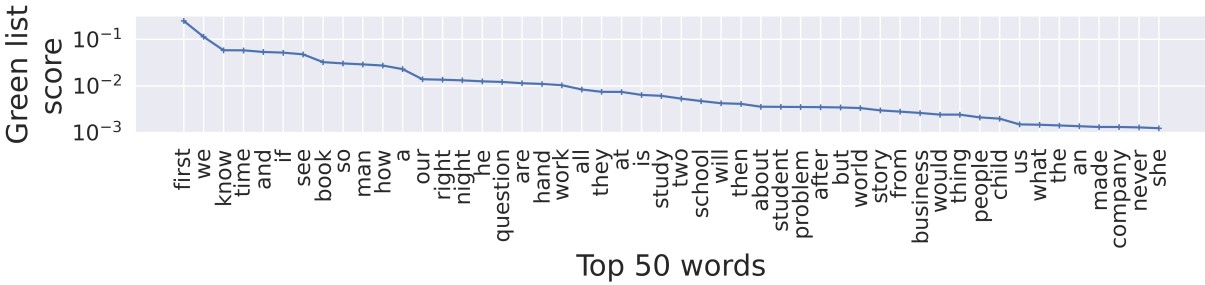

Figure 16: Inferred *green list score* for the token "the". The plot shows the top 50 words from our set of common words that are likely to be in the green list. The word "first" occurred $\sim 25\%$ of the time as suffix to "the".

generated by a human or an LLM. An *adversarial human* can compose derogatory watermarked texts in this fashion that are detected to be watermarked, which might cause reputational damage to the developers of the watermarked LLM. Therefore, it is important to study *spoofing attacks* to avoid such scenarios.

In watermarking, the prefix word $s^{(t-1)}$ determines the green list for selecting the word $s^{(t)}$. The attacker's objective is to compute a proxy of green lists for the $N$ most commonly used words in the vocabulary. We use a small value of $N = 181$ for our experiments. The attacker queries the watermarked OPT-1.3B Zhang et al. (2022) $10^6$ times to observe pair-wise token occurrences in its output to estimate *green list score* for the $N$ tokens. We find that inputting nonsense sentences composed of the $N$ common words encourages the LLM to output text mostly composed of these words. This makes the querying more efficient. A token with a high green list score for a prefix $s^{(t)}$ might be in its green list (see Figure 16). We build a tool that helps adversarial humans create watermarked sentences by providing them with proxy green list. In this manner, we can spoof watermarking models easily. See Table 15 for example sentences created by an adversarial human. Figure 7 shows that the performance of watermark-based detectors degrades significantly in the presence of paraphrasing and spoofing attacks, showing that they are not reliable.

| Human text | % tokens in green list | z-score | Detector output |
|---|---|---|---|
| the first thing you do will be the best thing you do. this is the reason why you do the first thing very well. if most of us did the first thing so well this world would be a lot better place. and it is a very well known fact. people from every place know this fact. time will prove this point to the all of us. as you get more money you will also get this fact like other people do. all of us should do the first thing very well. hence the first thing you do will be the best thing you do. | 42.6 | 4.36 | Watermarked |

Table 15: Proof-of-concept human-generated texts flagged as watermarked by the soft watermarking scheme. A sentence composed by an *adversarial human* contains 42.6% tokens from the green list. The z-test threshold for watermark detection is 4, the same as the default hyperparameter in Kirchenbauer et al. (2023a).

## D.2 Zero-Shot and Trained Detectors

We report the false positive rate fixed at a true positive rate of 90% and the true positive rate at a false positive rate of 1% in Table 16. The ROC curves before and after spoofing the detectors are provided in Figure 17. Our experiments demonstrate that most of these detection methods show a significant increase in

false positive rates at a fixed true positive rate of 90% after spoofing. After this naïve spoofing attack, the true positive rate at a false positive rate of 1% and AUROC scores of these detectors drop significantly.

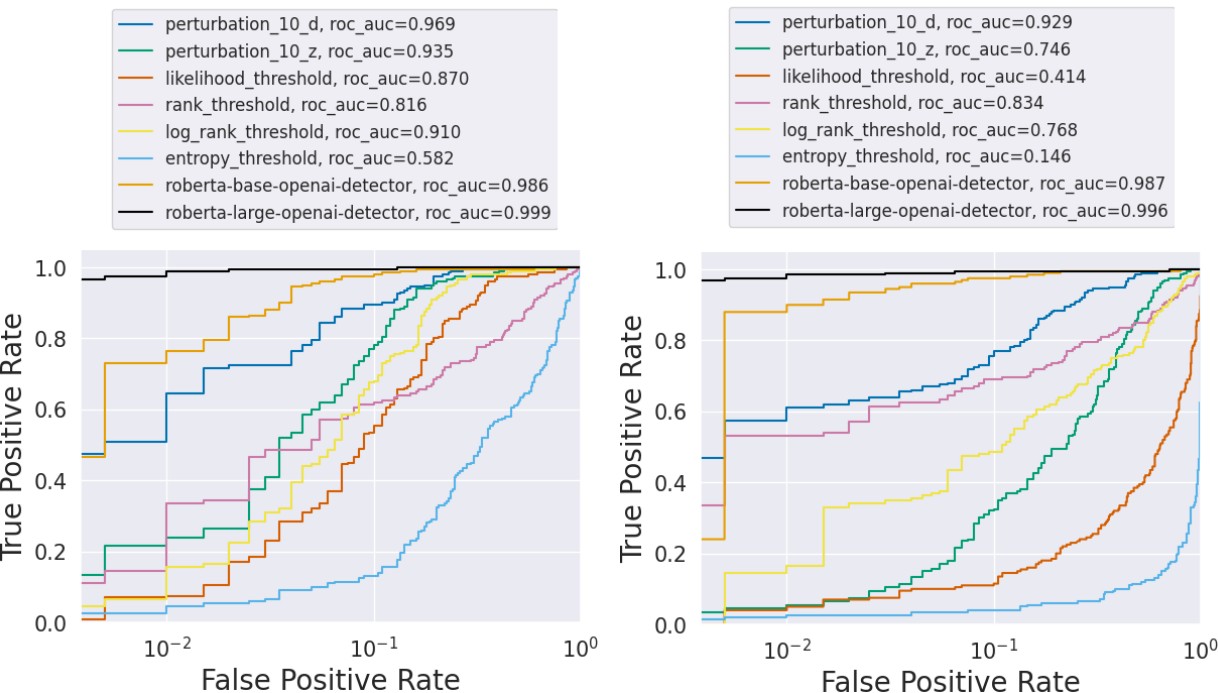

Figure 17: ROC curves before (left) and after (right) spoofing attack (§ 3). Most detectors exhibit quality degradation after our spoofing attack.

| Detection Methods | T@F | F@T |
|---|---|---|
| Entropy threshold (Gehrmann et al., 2019) | **0.025** (0.045) | **0.995** (0.845) |
| Likelihood threshold (Solaiman et al., 2019) | **0.050** (0.075) | **0.995** (0.310) |
| Logrank threshold | 0.165 (0.155) | **0.690** (0.190) |
| Rank threshold (Gehrmann et al., 2019) | 0.530 (0.335) | **0.655** (0.590) |
| Roberta (base) OpenAI detector (OpenAI, 2019) | 0.900 (0.765) | 0.010 (0.035) |
| Roberta (large) OpenAI detector (OpenAI, 2019) | **0.985** (0.990) | 0.000 (0.000) |
| DetectGPT (Mitchell et al., 2023) | **0.055** (0.240) | **0.555** (0.145) |

Table 16: True positive rates at 1% false positive rate (T@F) and false positive rates at 90% true positive rate (F@T) after (before the attack in parentheses) the spoofing attack described in §3. Bolded numbers show successful attacks where T@F decreases, or F@T increases after spoofing.

# E   Conclusion

In this paper, we stress-test the performance of four different classes of detectors, including watermarking, neural net, zero-shot, and retrieval-based detectors in the presence of an attacker. We develop a strong evasion attack called *recursive paraphrasing* that can break recently proposed watermarking and retrieval-based detectors. We use MTurk human studies and other automated metrics to quantify the degradation in text quality after our attack. We also show that adversaries can spoof these detectors to increase their type-I errors and cause reputational damage to LLM developers. Finally, we establish a theoretical connection between the AUROC of the best possible detector and the TV distance between human and AI-text distributions that can be used to study the fundamental hardness of the reliable detection problem for more advanced LLMs.

In the future, attackers could adversarially train LLMs to explicitly mimic a specific group of people to minimize TV distances based on our theory to evade detection easily. It might be interesting to see more work on this aspect. Though the existing paraphrasers we use are powerful, they might not perform as well in specific technical domains such as clinical text data. However, stronger paraphrasers in the future might overcome these issues. By showing empirical evidence on larger models having smaller TV distance estimates, we hypothesize that reliable detection would get harder as LLMs get more powerful.

A detector should ideally be helpful in reliably flagging AI-generated texts to prevent the misuse of LLMs. However, the cost of misidentification by a detector can be huge. If the false positive rate of the detector is not low enough, humans (e.g., students) could be falsely accused of AI plagiarism. Moreover, a disparaging passage falsely detected to be AI-generated could affect the reputation of the LLM's developers. As a result, the practical applications of AI-text detectors can become unreliable and invalid. Security methods need not be foolproof. However, we need to make sure that it is not an easy task for an attacker to break these security defenses. Thus, stress testing current and future detectors can be vital to avoid creating a false sense of security.

