# OpenReview forum: "Can AI-Generated Text be Reliably Detected? Stress Testing AI Text Detectors Under Various Attacks"
_TMLR — Accepted by TMLR_

### Review · Reviewer_thTT · 2024-11-08

**Summary Of Contributions:**

This paper studies the problem of detecting AI-generated text. With the assumption of the existence of a malicious adversary, the authors stress-test the robustness and reliability of existing detectors, including watermarked LLM, zero-shot and trained detector, and retrieval-based detector. The experiments demonstrate (almost) all above detectors are non-robust to the proposed recursive paraphrasing attack, and further, the adversary can develop a type of spoofing attack to mislead the classifier to misclassify human-written text as AI-generated. Finally, a theoretical bound on the AUROC of any detectors is provided based on the total variation distance between the distributions one would distinguish.

**Audience:**

Yes

**Claims And Evidence:**

Yes

**Requested Changes:**

As demonstrated above, I think it would be beneficial to incorporate the following modifications into the draft.

- In section 2.2. What is the averaged length of the tested contexts in the picked QA questions? If it is significantly less than 300 tokens, then additional utility test with longer contexts (for example, in-context MMLU evaluation) could serve as a stronger evidence to support the claim.

- In section 2.3, it would be nice to add a discussion on the abnormal behavior of the entropy threshold detector.

- In section 2.4, the trained detector still performs reasonably well after the recursive paraphrasing attack is launched. If possible, additional results with trained detectors on larger target LLMs would be beneficial. Otherwise, it is insufficient to claim AI-generated content is not detectable.

**Strengths And Weaknesses:**

**Strengths.**

- The paper is well written, and the studied problem is timely and intriguing.
- The strong performance of the recursive paraphrasing attack advocates for rethinking the effectiveness of current AI-generated content detectors.
- The proposed spoofing attack is novel and the perspective is noteworthy.

**Weaknesses.**

My concerns on weaknesses are mainly concentrated in section 2.

- In section 2.2, when testing the quality of the paraphrases, both human evaluation and QA benchmark are considered. However, the performance change on the studied QA benchmark may be insufficient to support the claim, since the context passages in SQuAD-v2 may be varying in length, and thus does not cover the cases of paraphrasing long passages. A numerical description on the length of these contexts could be useful - otherwise more evidence is necessary, e.g., using few-shot in-context tasks such as MMLU evaluation.

- In section 2.3, Figure 5, though almost all detectors suffer from a performance degradation after the paraphrasing attack, the entropy threshold detector performs **better** after the attack is launched. This is an intriguing observation, but there is no discussion on this specific behavior. A discussion on this will be attractive, since it may motivate better detectors in practice against the adversary.

- In section 2.4, with the T5-based paraphraser, the **trained detector**, roberta-large-openai, still performs reasonably well. Thus, it may be insufficient to claim AI-generated content is not detectable, and is beneficial to highlight a trained detector's robustness against the recursive paraphrasing attack.

---

> ### Author Response · Authors · 2024-12-05
> **Response to Reviewer thTT**
>
> We thank the reviewer for their insightful comments. We address their comments below. We added revisions to our manuscript based on the reviewer's comments in brown text.
>
> > In section 2.2. What is the averaged length of the tested contexts in the picked QA questions?
>
> Thanks for this comment. We add this detail to the revised manuscript. For the automated study, we select SQuAD-v2 data points that have context texts that are at least 300 tokens in length. The mean length of context text is 328 $\pm$ 28 tokens, which is similar to the length of text passages we use for our attack experiments for detectors.
>
> > it would be nice to add a discussion on the abnormal behavior of the entropy threshold detector.
>
> As noted by the reviewer, the detection score of the entropy threshold detector increases with more rounds of paraphrasing. We hypothesize that this might be due to memorization in LLMs. LLMs are trained on human-written texts, and for this reason, they might have low entropy scores on human-written samples we use in our experiments due to memorization. Therefore, the entropy detector might have poor detection scores before the paraphrasing attack. However, after paraphrasing with a different AI model, the entropy scores for these human-written samples might increase, improving the detection scores. In spite of this, we note that the entropy threshold detector has poor detection rates before and after the attack.
>
> > If possible, additional results with trained detectors on larger target LLMs would be beneficial.
>
> Thanks for the comment. We add results with target LLM as Llama-2-13b for various trained detectors in the revised manuscript Appendix A.1. Please find the results in the table below.
>
> | TPR at 1% FPR | MAGE | Longformer | roberta-base | roberta-large |
> |---------------|------|------------|--------------|---------------|
> | No attack     | 0.457| 0.772      | 0.672        | 0.648         |
> | pp1           | 0.405| 0.476      | 0.404        | 0.6           |
> | pp2           | 0.4  | 0.454      | 0.316        | 0.556         |
> | pp3           | 0.421| 0.424      | 0.304        | 0.56          |
> | pp4           | 0.427| 0.436      | 0.296        | 0.504         |
> | pp5           | 0.421| 0.432      | 0.304        | 0.524         |

---

### Review · Reviewer_ZpWr · 2024-11-15

**Summary Of Contributions:**

The paper considers an important problem of detecting AI-generated texts. The main contributions are:
- Theoretical analysis on feasibility of constructing AI-generated text detectors;
- Experimental analysis on performances of various AI-generated text detectors under certain types of attacks;
- A new setup of paraphrasing attack which works well on many AI-generated text detectors.

**Audience:**

Yes

**Broader Impact Concerns:**

-

**Claims And Evidence:**

Yes

**Requested Changes:**

- The most important issue for me seems to be a choice of LLMs for testing detectors (GPT-2 only or GPT-2/OTB). I would appreciate adding experiments on newer models such as versions of LLama/GPT.
- It would also be nice to see other types of attacks and detectors added.

**Strengths And Weaknesses:**

Strengths:
- The topic considered in the paper is important and of interest to the community;
- A wide range of detectors is considered;
- Proposed recursive paraphrasing attack is simple and works quite well on wide range of detectors;
- Experiments are thorough;
- Nice theoretical findings which support the general intuition on the hardness of the problem.
- Interesting insights on how human-generated text can be classified as AI-generated, and spoofing attacks.

Overall, work presented in the paper is quite comprehensive and provides valuable insights on the question of reliability of AI-generated text detectors.

Weaknesses, questions and comments:
- Choice of LLMs for testing detectors is scarce: OTB-13B/OTB-1.3B and variants of GPT-2 are considered for testing watermark-based detectors, and only GPT-2 is considered for non-watermark-based detectors. GPT-2 is quite outdated model by now, and it would make more sense to have experiments on newer, stronger models, such as Llama/GPT versions, as well as their instruction-tuned variants;
- There are other types of AI-generated text detectors. For example, intristic-dimension based (https://proceedings.neurips.cc/paper_files/paper/2023/hash/7baa48bc166aa2013d78cbdc15010530-Abstract-Conference.html), topological-based (https://arxiv.org/abs/2109.04825), or based on modified embeddings (https://arxiv.org/abs/2410.08113). Also GPTZero (https://gptzero.me) is often used as one of the baselines of detectors. It would be interesting to test proposed attack on these detectors;
- There are other types of attacks on AI-generated text detectors, such as introducing grammatical mistakes, SICO (https://arxiv.org/pdf/2305.10847). It would be interesting to compare the performance of proposed recursive paraphrasing attack to those;
- Results show that RoBERTa-based detector is quite robust against proposed recursive paraphrasing attack. It would be nice to understand why that is so, which may lead to insights on how to build a robust AI-generated text detector;
- A statement on spoofing retrieval-based detectors seems to be shallow, as it is claimed by authors that any target LLM would store a paraphrased copy of the text S' in its database, which is then used for retrieval purposes. It is possible to have a retrieval-based detector which would not store any produced result in its database, so the claim of universal weakness of retrieval-based detectors seems to be incorrect;

---

> ### Author Response · Authors · 2024-12-05
> **Response to Reviewer ZpWr**
>
> We thank the reviewer for their insightful comments. We address their comments below. We have added revisions to our manuscript based on the reviewer's comments in brown text.
>
> > it would make more sense to have experiments on newer, stronger models, such as Llama/GPT versions
>
> Thanks for this comment. We add new results with Llama-2-13b as the target model in our revised Appendix A.1. Our new experiment results are consistent with our previous results. We also provide the results in the table below:
>
> | Detector | perturbation_1_d | perturbation_1_z | perturbation_10_d | perturbation_10_z | likelihood_threshold | rank_threshold | log_rank_threshold | entropy_threshold | MAGE | Longformer | roberta-base | roberta-large | Kuditipudi |
> |---------------|------------------|------------------|-------------------|-------------------|----------------------|-----------------|---------------------|-------------------|------------|------------------------------|------------------------------|-------------------------------|----------------------------|
> | No attack | 0.32 | 0.32 | 0.612 | 0.048 | 0.992 | 0.148 | 0.98 | 0.0 | 0.457 | 0.772 | 0.672 | 0.648 | 0.988 |
> | pp1 | 0.1 | 0.1 | 0.204 | 0.0 | 0.652 | 0.1 | 0.564 | 0.104 | 0.405 | 0.476 | 0.404 | 0.6 | 0.532 |
> | pp2 | 0.142 | 0.142 | 0.124 | 0.02 | 0.536 | 0.076 | 0.444 | 0.116 | 0.4 | 0.454 | 0.316 | 0.556 | 0.356 |
> | pp3 | 0.04 | 0.04 | 0.068 | 0.036 | 0.516 | 0.076 | 0.432 | 0.104 | 0.421 | 0.424 | 0.304 | 0.56 | 0.322 |
> | pp4 | 0.04 | 0.04 | 0.052 | 0.02 | 0.492 | 0.08 | 0.412 | 0.12 | 0.427 | 0.436 | 0.296 | 0.504 | 0.292 |
> | pp5 | 0.068 | 0.068 | 0.06 | 0.0 | 0.476 | 0.08 | 0.388 | 0.12 | 0.421 | 0.432 | 0.304 | 0.524 | 0.264 |
>
>
> > There are other types of AI-generated text detectors.
>
> Thanks for the comment. We add more results on Longformer and MAGE-based detectors [Li et al.] (https://aclanthology.org/2024.acl-long.3/) as shown in the table above in our revised manuscript. We find similar observations of performance degradation after the attack with these detectors.
>
> > There are other types of attacks on AI-generated text detectors, such as introducing grammatical mistakes, SICO (https://arxiv.org/pdf/2305.10847).
>
> Thanks for the comment. Introducing grammatical errors would degrade the text quality even further, and therefore, we resort to adding this to our attack since we also focus on maintaining text quality after the attack.
> SICO employs in-context optimization through iterative generation of word- or sentence-level substitutions using a proxy AI text detector to guide the substitutions. This positions SICO as an adversarial algorithm for evading text detection. In contrast, our approach focuses on non-adversarial iterative or recursive text paraphrasing attacks. We have cited this reference and added an explanation about it in the revised draft.
> We also share the following information with the permission of AE: we would like to note that SICO explicitly cites an arXiv version of our paper as prior work since we are the first to explore evasion attacks on a variety of detectors.
>
>
> > Results show that RoBERTa-based detector is quite robust against proposed recursive paraphrasing attack.
>
> Thanks for this comment. We add more trained detectors (Longformer and MAGE-based detectors [https://aclanthology.org/2024.acl-long.3/]) to our study. Though their performance degrades after each round of paraphrasing, we note that they are more robust to paraphrase attacks than the other detectors we study. We hypothesize that this might be due to these detectors being trained on human-written samples we use for our study. For example, the MAGE dataset includes passages from the XSum dataset that we use. Recent work by Gameiro et al. (https://arxiv.org/abs/2409.03291) also argues that while trained detectors can generalize better to unseen LLMs, they may overfit to this training distribution of human text. They also show that some of these detectors fail to generalize to out-of-distribution human-written text. This is an aspect that we do not consider in our work, but would still make these detectors unreliable for real-world applications.
>
>
> >  It is possible to have a retrieval-based detector which would not store any produced result in its database
>
> The retrieval-based detector proposed by Krishna et al. that we consider stores the AI outputs (or its neural representations) in a database. Therefore, our spoofing attack is successful with this detector as shown in our experiments.

---

### Review · Reviewer_cJXn · 2024-11-18

**Summary Of Contributions:**

The paper evaluates the effectiveness of paraphrasing and spoofing attacks on various existing methods for detecting AI-generated text. Specifically, the paper considers attacking the watermarking scheme of [1] and various post-hoc detection methods.

The paper proposes a method for recursively paraphrasing AI-generated text by repeatedly prompting a different language model (i.e., different from the model used to generate the text in the first place) to rewrite the text. The paper demonstrates the effectiveness of this attack via experiments on several different tasks/datasets. Also, through both human and machine evaluations, the paper argues that the rewriting does not degrade the quality of the text.

The paper also considers spoofing watermarked text—again using the watermark of [1]—and spoofing text that fools a retrieval-based detection method. In the former case, the paper proposes a method for estimating the green list tokens of [1]'s watermark. In the later case, the paper prompts the target model to paraphrase some piece of text, which by construction fools the retrieval-based detection method.

Finally, the paper proves that it is impossible to accurately test between two distributions that are similar in total variation distance.

[1] https://arxiv.org/abs/2301.10226
[2] https://arxiv.org/abs/2311.04378
[3] https://arxiv.org/abs/2307.15593
[4] https://arxiv.org/abs/2306.17439
[5] https://arxiv.org/abs/2312.04469

**Audience:**

Yes

**Claims And Evidence:**

No

**Requested Changes:**

The paper should do a detailed comparison to prior work on evading watermark detection by recursive paraphrasing [2] and spoofing watermarks [5]. Also, resources permitting, the paper should ideally run a blind human evaluation to determine whether there is a significant difference in quality between the original AI-generated text versus the output of their paraphrasing attacks (see Weaknesses section for details).

I have currently marked "No" under "Claims and Evidence" mainly because of the way in which the human evaluation was done and because the paper does not adequately compare to prior work. Both of these issues seem fixable, and overall the paper is reasonably well done.

[1] https://arxiv.org/abs/2301.10226
[2] https://arxiv.org/abs/2311.04378
[3] https://arxiv.org/abs/2307.15593
[4] https://arxiv.org/abs/2306.17439
[5] https://arxiv.org/abs/2312.04469

**Strengths And Weaknesses:**

The main strength of the paper is that it evaluates the effectiveness of its attacks on a diverse set of performant post-hoc detection methods and a number of different tasks/datasets. The paper also goes to the effort of conducting human evaluations to show that its attacks do not degrade text quality. Finally, the spoofing attack on the retrieval-based detection method is clever.

In comparison, one major weakness of the paper is that it only evaluates one watermarking method, i.e., that of Kirchenbauer et al. Furthermore, the paper does not compare their results to prior work [2] which also considers breaking various watermarking schemes by recursive paraphrasing. This prior work [2] evaluates not just the scheme of [1] but also [3,4]. Notably, unlike Kirchenbauer et al. these other schemes do not change the sampling distribution over text; thus, the total variation distance lower bound does not directly apply. Also, for the same reason, prior work [5] which also studies spoofing various watermarking schemes found that it is comparatively harder to spoof [3,4] than [1].

Instead of providing human evaluators with the original AI-generated text and asking them to rate whether the paraphrased version is of similar quality, a more convincing human evaluation protocol would be to have evaluators blindly rate the quality of the original versus paraphrased versions and see if there is a statistically significant difference. Also, in some cases the paper uses a stronger model (e.g., LLama-2-7B-Chat to paraphrase text generated by a weaker model (e.g, OPT-13B) to evade detection. At least in the case of watermarking, a more realistic attack (and arguably a more compelling result) would be to use a weaker model to paraphrase text generated by a stronger model, as done by [2].

[1] https://arxiv.org/abs/2301.10226
[2] https://arxiv.org/abs/2311.04378
[3] https://arxiv.org/abs/2307.15593
[4] https://arxiv.org/abs/2306.17439
[5] https://arxiv.org/abs/2312.04469

---

> ### Author Response · Authors · 2024-12-05
> **Response to Reviewer cJXn**
>
> We thank the reviewer for their insightful comments. We address their comments below. We have added revisions to our manuscript based on the reviewer's comments in brown text.
>
> > In comparison, one major weakness of the paper is that it only evaluates one watermarking method, i.e., that of Kirchenbauer et al.
>
> We add new results with the watermarking scheme from Kuditipudi et al. (our revised Appendix A.1). We find that recursive paraphrasing can degrade the detection of even distortion-free watermarking schemes.
> With Llama-2-13b as the target model and Llama-2-7b as the attacker’s paraphrasing model, we get the following detection scores (TPR@1%FPR).
>
> |No attack|pp1|pp2|pp3|pp4|pp5|
> |-------------|-----|-----|-----|-----|-----|
> |98.8|53.2|35.6|32.2|29.2|26.4|
>
>
> > The paper should do a detailed comparison to prior work on evading watermark detection by recursive paraphrasing [2] and spoofing watermarks [5].
>
> Recursive paraphrasing:
>
> As mentioned in [2] (their Appendix A), Zhang et al. have a different attack objective when compared to our attack. Their attack is performed to maintain the quality of the text alone and not semantic similarity. Hence, their attack can change the original content significantly. Therefore, we do not believe it is an appropriate baseline for our attack with theirs.
>
> We also share the following information with the permission of AE: we would like to note that Zhang et al. explicitly cites an arXiv version of our paper as prior work since we are the first to explore evasion attacks on a variety of detectors. We hope this additional information addresses the reviewer's comment.
>
> Spoofing:
>
> Our watermark spoofing method approximates the next-token distribution by analyzing token-pair distributions in watermarked text samples. Gu et al. [5] build upon this idea by employing watermarked data distillation to train a student model to replicate the next-token distribution. We have added a description of the Gu et al. [5] work to the paper.
>
> We also share the following information with the permission of AE: We note that the work by Gu et al. [5] builds on an earlier arXiv version of our work. Thus, a direct comparison would not be entirely appropriate. Our work was the first one to introduce the spoofing attack framework, which motivated several follow-up works in this space. We hope this additional information addresses the reviewer's comment.
>
>
> > Also, in some cases the paper uses a stronger model (e.g., LLama-2-7B-Chat to paraphrase text generated by a weaker model (e.g, OPT-13B) to evade detection.
>
> Thanks for this comment. We add new experiments where the target model (Llama-2-13B) is larger and more powerful than the paraphrasing model (Llama-2-7B) in our revised Appendix A.1.
> Our observations with the new experiments are consistent with our previous results, i.e., the performance detectors get worse with AI paraphrasing. We also add the results below:
>
>
> | Detector | perturbation_1_d | perturbation_1_z | perturbation_10_d | perturbation_10_z | likelihood_threshold | rank_threshold | log_rank_threshold | entropy_threshold | MAGE | Longformer | roberta-base | roberta-large | Kuditipudi |
> |---------------|------------------|------------------|-------------------|-------------------|----------------------|-----------------|---------------------|-------------------|------------|------------------------------|------------------------------|-------------------------------|----------------------------|
> | No attack | 0.32 | 0.32 | 0.612 | 0.048 | 0.992 | 0.148 | 0.98 | 0.0 | 0.457 | 0.772 | 0.672 | 0.648 | 0.988 |
> | pp1 | 0.1 | 0.1 | 0.204 | 0.0 | 0.652 | 0.1 | 0.564 | 0.104 | 0.405 | 0.476 | 0.404 | 0.6 | 0.532 |
> | pp2 | 0.142 | 0.142 | 0.124 | 0.02 | 0.536 | 0.076 | 0.444 | 0.116 | 0.4 | 0.454 | 0.316 | 0.556 | 0.356 |
> | pp3 | 0.04 | 0.04 | 0.068 | 0.036 | 0.516 | 0.076 | 0.432 | 0.104 | 0.421 | 0.424 | 0.304 | 0.56 | 0.322 |
> | pp4 | 0.04 | 0.04 | 0.052 | 0.02 | 0.492 | 0.08 | 0.412 | 0.12 | 0.427 | 0.436 | 0.296 | 0.504 | 0.292 |
> | pp5 | 0.068 | 0.068 | 0.06 | 0.0 | 0.476 | 0.08 | 0.388 | 0.12 | 0.421 | 0.432 | 0.304 | 0.524 | 0.264 |
>
>
>
>
>
> > Also, resources permitting, the paper should ideally run a blind human evaluation to determine whether there is a significant difference in quality between the original AI-generated text versus the output of their paraphrasing attacks
>
> Thanks for the comment. Unfortunately, we do not have the resources to add further human study to this. However, we would like to note that we use Likert scales with well-defined rating instructions for our human study based on other well-established works on paraphrase evaluations Michail et al. (https://arxiv.org/pdf/2409.12060) and Krishna et al. (https://arxiv.org/abs/2303.13408).

---

### Decision · Action_Editor_3B3q · 2024-12-24

**Recommendation:** Accept as is

**Comment:**

This is a good and solid contribution, and the reviewers are united in their recommendations of acceptance.

**Audience:**

Audience is general trustworthy AI community

**Claims And Evidence:**

This paper looks at a problem of general interest to the AI safety/trustworthy AI community, and makes solid contributions that are backed up by evidence. Hence I recommend acceptance.